# Noether Networks:
# Meta-Learning Useful Conserved Quantities

**Ferran Alet**[*1], **Dylan Doblar**[*1], **Allan Zhou**[2],
**Joshua Tenenbaum**[1]**, Kenji Kawaguchi**[3]**, Chelsea Finn**[2]
[1]MIT, [2]Stanford University, [3]National University of Singapore
{alet,ddoblar}@mit.edu

## Abstract

Progress in machine learning (ML) stems from a combination of data availability, computational resources, and an appropriate encoding of inductive biases. Useful biases often exploit symmetries in the prediction problem, such as convolutional networks relying on translation equivariance. Automatically discovering these useful symmetries holds the potential to greatly improve the performance of ML systems, but still remains a challenge. In this work, we focus on sequential prediction problems and take inspiration from Noether's theorem to reduce the problem of finding inductive biases to meta-learning useful conserved quantities. We propose Noether Networks: a new type of architecture where a meta-learned conservation loss is optimized inside the prediction function. We show, theoretically and experimentally, that Noether Networks improve prediction quality, providing a general framework for discovering inductive biases in sequential problems.

## 1 Introduction

The clever use of inductive biases to exploit symmetries has been at the heart of many landmark achievements in machine learning, such as translation invariance in CNN image classification [35], permutation invariance in Graph Neural Networks [52] for drug design [57], or roto-translational equivariance in SE3-transformers [25] for protein structure prediction [33]. However, for data distributions of interest, there may be exploitable symmetries that are either unknown or difficult to describe with code. Progress has been made in automatically discovering symmetries for finite groups [74], but meta-learning and exploiting general continuous symmetries has presented a major challenge. In part, this is because symmetries describe the effect of counterfactuals about perturbations to the data, which are not directly observable.

In this work, we propose to exploit symmetries in sequential prediction problems indirectly. We take inspiration from Noether's theorem [44], which loosely states the following:

> *For every continuous symmetry property of a dynamical system,*
> *there is a corresponding quantity whose value is conserved in time.*

For example, consider a system of planets interacting via gravity: this system is translation invariant in all three cardinal directions (i.e. translating the entire system in the x,y, or z axis conserves the laws of motion). Noether's theorem asserts there must be a conserved quantity for each of these symmetries; in this case, linear momentum. Similarly, the system has a time-invariance (i.e. the laws of motion are the same today as they will be tomorrow). In this case, the corresponding conserved quantity is the total energy of the system.

---

[*]Equal contribution. Our code is publicly available at https://lis.csail.mit.edu/noether.

35th Conference on Neural Information Processing Systems (NeurIPS 2021).

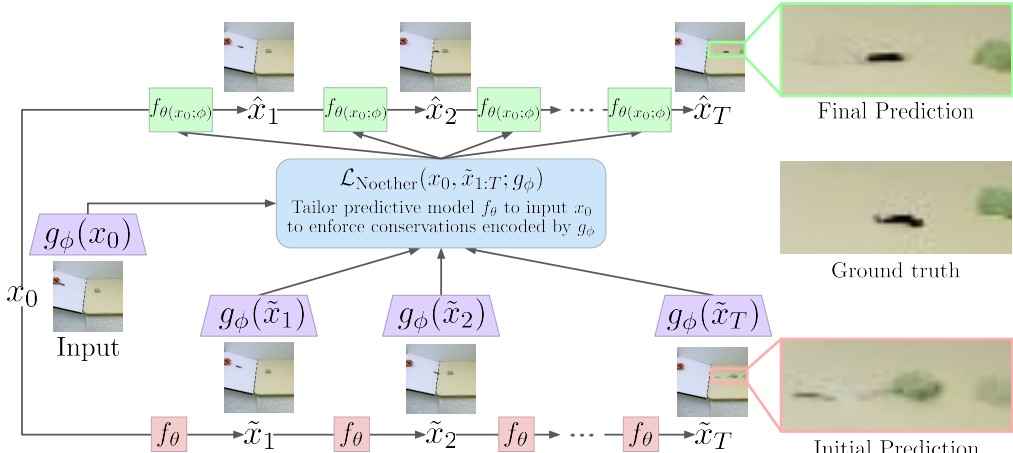

Figure 1: Noether Networks enforce conservation laws, meta-learned by $g_\phi$, in sequential predictions made by $f_\theta$, which is tailored to the input $x_0$ to produce final predictions $\hat{x}_{1:T}$. Imposing these meta-learned inductive biases improves video prediction quality with objects sliding down a ramp.

Inspired by this equivalence, we propose that approximate conservation laws are a powerful paradigm for meta-learning useful inductive biases in sequential prediction problems. Whereas symmetries are difficult to discover because they are global properties linked to counterfactuals about unobserved perturbations of the data, conserved quantities can be directly observed in the true data. This provides an immediate signal for machine learning algorithms to exploit.

Our approach involves meta-learning a parametric conservation loss function which is useful to the prediction task. We leverage the *tailoring* framework [4], which proposes to encode inductive biases by fine-tuning neural networks with hand-designed unsupervised losses inside the prediction function. Whereas traditional auxiliary losses are added to the task loss during training, tailoring losses are optimized inside the prediction function both during training and testing, customizing the model to each individual query. In doing so, we ensure there is no generalization gap for the conservation loss. We propose to meta-learn the self-supervised loss function and parameterize it in the form of a conservation loss; i.e. $\mathcal{L}(x_0, \tilde{x}_{1:T}; g_\phi) = \sum_{t=1}^{T} |g_\phi(x_0) - g_\phi(\tilde{x}_t)|^2$. This conservation form encodes a meta-inductive bias (inductive bias over inductive biases) which narrows the search space exponentially (in $T$) and simplifies the parameterization. Figure 1 demonstrates the Noether Network pipeline for a video prediction task.

The main contribution of this paper is the Noether Network, an architecture class and algorithm that can automatically learn its own inductive biases in the form of meta-learned conservation loss functions. We theoretically characterize the advantages of such conservation laws as effective regularizers that constrain the learning to lower-dimensional output manifolds, lowering the generalization gap. Empirically, we first find that, when the meta-learned conservation loss takes the form of a synthesized program, Noether Networks recover known conservation laws from raw physical data. Second, we find that Noether Networks can learn conservation losses on raw videos, modestly improving the generalization of a video prediction model, especially on longer-term predictions.

## 2 Theoretical Advantages of Enforcing Conservation Laws

In this section, we demonstrate principled advantages of enforcing conservation laws of the form $g_\phi(f_\theta(x)) = g_\phi(x)$ by considering a special case where preimages under $g_\phi$ form affine subspaces.

Let input $x$ and target $y$ be $x, y \in \mathbb{R}^d$, and let the Noether embedding be $g_\phi : \mathbb{R}^d \to \mathcal{P}$ where $\mathcal{P} = \{g_\phi(x) : x \in \mathbb{R}^d\}$. We consider a restricted class of models, parameterized by $\theta \in \Theta$, of the form $f_\theta(x) = x + v_\theta$ for $v_\theta \in \mathbb{R}^d$ such that for all $x$, the preimage of $g_\phi$ is $g_\phi^{-1}[\{g_\phi(x)\}] = \{x + Az : z \in \mathbb{R}^m\}$, $A \in \mathbb{R}^{d \times m}$. Here, $m \leq d$ is the dimensionality of the preimages of $g_\phi$. We denote by $C$ the smallest upper bound on the loss value as $\mathcal{L}(f_\theta(x), y) \leq C$ (for all $x, y$ and $\theta$).

Define $\psi(v) = \mathbb{E}_{x,y}[\mathcal{L}(f(x,v), y)] - \frac{1}{n}\sum_{i=1}^{n} \mathcal{L}(f(x_i, v), y_i)$, with Lipschitz constant $\zeta$. Therefore, $\zeta \geq 0$ is the smallest real number such that, for all $v$ and $v'$ in $\mathcal{V}$, $|\psi(v) - \psi(v')| \leq \zeta \|v - v'\|_2$, where $\mathcal{V} = \{v_\theta \in \mathbb{R}^d : \theta \in \Theta\}$. Finally, we define $R = \sup_{\theta \in \Theta} \|v_\theta\|_2$, and the generalization gap by

$$G(\theta) = \mathbb{E}_{x,y}[\mathcal{L}(f_\theta(x), y)] - \frac{1}{n}\sum_{i=1}^{n} \mathcal{L}(f_\theta(x_i), y_i).$$

Theorem 1 shows that enforcing conservation laws of $g_\phi(f_\theta(x)) = g_\phi(x)$ is advantageous when the dimension of the preimages under $g_\phi$ is less than the dimension of $x$; that is, when $m < d$.

**Theorem 1.** *Let $\rho \in \mathbb{N}^+$. Then, for any $\delta > 0$, with probability at least $1 - \delta$ over an iid draw of $n$ examples $((x_i, y_i))_{i=1}^{n}$, the following holds for all $\theta \in \Theta$:*

$$G(\theta) \leq C\sqrt{\frac{\xi \ln(\max(\sqrt{\xi}, 1)) + \xi \ln(2R(\zeta^{1-1/\rho})\sqrt{n}) + \ln(1/\delta)}{2n}} + \mathbb{1}\{\xi \geq 1\}\sqrt{\frac{\zeta^{2/\rho}}{n}}. \qquad (1)$$

*where $\xi = m$ if $g_\phi(f_\theta(x)) = g_\phi(x)$ for any $x \in \mathbb{R}^d$ and $\theta \in \Theta$, and $\xi = d$ otherwise.*

The proof is presented in Appendix A. In Theorem 1, when we enforce the conservation laws, $d$ is replaced by $m$, the dimension of the preimage. We now discuss various cases for the values of $m$:

- (Case of $m = d$) Let us consider an extreme scenario where the function $g_\phi$ maps all $x \in \mathbb{R}^d$ to one single point. In this scenario, the dimensionality of preimages under $g_\phi$ is maximized as $m = d$. Accordingly, the bounds with and without enforcing the conservation laws become the same. Indeed, in this scenario, the conservation laws of $g_\phi(f_\theta(x)) = g_\phi(x)$ give us no information, because they always hold for all $x \in \mathbb{R}^d$, even without imposing them.

- (Case of $m = 0$) Let us consider another extreme scenario where the function $g_\phi$ is invertible. In this scenario, the dimensionality of preimages under $g_\phi$ is zero as $m = 0$. Thus, imposing the condition of $g_\phi(f_\theta(x)) = g_\phi(x)$ makes the bound in Theorem 1 to be very small. Indeed, in this scenario, the condition of $g_\phi(f_\theta(x)) = g_\phi(x)$ implies that $f_\theta(x) = x$: i.e., $x$ is not moving, and thus it is easy to generalize.

- (Case of $0 < m < d$) From the above two cases, we can see that the benefit of enforcing conservation laws comes from more practical cases in-between these, with $0 < m < d$.

In Theorem 1, the function $g_\phi$ can differ from the true function $g_\phi^*$ underlying the system. This is because we analyze a standard generalization gap: i.e., the difference between the expected loss and the training loss. The cost of not using the true $g_\phi^*$ is captured in the training loss; i.e., the training loss can be large with the function $g_\phi$ that differs significantly from the true $g_\phi^*$. Even in this case, the generalization gap can be small. For example, in the case of $m = 0$, the generalization bound is small, whereas the training loss will be large unless $x_{t+1} = x_t$. Therefore, our analysis gives us the insight on the trade-off between the training loss and the dimensionality of preimages under $g_\phi$.

## 3 Noether Networks

**Leveraging tailoring to encode inductive biases.** We perform a prediction-time optimization to encourage outputs to follow conservation laws using the tailoring framework [4]. Tailoring encodes inductive biases in the form of unsupervised losses optimized inside the prediction function. In doing so, tailoring fine-tunes the model to each query to ensure that it minimizes the unsupervised loss for that query. For example, we may optimize for energy conservation in a physics prediction problem. In meta-tailoring, we train the model to do well on the task loss after the tailoring step has fine-tuned its parameters. In contrast to auxiliary losses, which would optimise the conservation loss only for the training points, tailoring allows us to ensure conservation at test time. Since we aim to build in the conservation in the architecture, we want to ensure it is also satisfied for unseen test samples. Another advantage of tailoring losses is that they are easier to meta-learn. Auxiliary losses are pooled over all examples and training epochs and their effect is only known at validation/test time. We would need to use implicit gradients [39, 47] to know their eventual effect on the final weights at the end of training. With tailoring, we can directly measure the effect of the meta-learned update on the same sample.

A limitation of tailoring framework is that the tailoring loss must be user-specified. This is acceptable in domains where the desired inductive bias is both known and easily encoded, but problematic in general — we address this issue with Noether Networks. Our approach can be seen as a generalization of tailoring where the unsupervised loss is meta-learned and takes the form of a conservation loss.

**Noether Networks: discovering *useful* conservation laws.** We build on the paradigm of conservation laws; i.e. quantities that are conserved over time. This has the following two challenges:

1. Real data often has noise, breaking any exact conservation law. Moreover, many conservation laws are only approximately satisfied given only partial information of the system. For example, conservation of energy is not fully satisfied in real dissipative systems and estimating conservation of mass from pixels is bound to be inexact in the case of occlusions.

2. Optimizing for conservation alone can lead to trivial quantities, such as predicting a constant value $g_\phi(x) = C$ independent of $x$.

We search for *useful* conservation laws, whose (approximate) enforcing brings us closer to the true data manifold for the current sequence. Note that a useful conserved quantity doesn't need to be exactly conserved in the training data to improve prediction performance. We only need its conservation to be informative of the data manifold. This allows us to discover conservation laws even in imperfect, noisy environments. Conversely, tailoring with a trivial conserved quantity cannot improve the final predictions (see App. B for a detailed explanation). Finally, viewing conserved quantities as useful inductive biases aligns well with their use, since we often care about biases only insofar as they help improve task performance.

The search for such useful conserved quantities can take many forms. In this work, we focus on two: first, in the beginning of section 4.1, we use a combination of program synthesis and gradient descent to generate a large, but finite, number of candidate parametric formulas for physical system prediction. We then try meta-tailoring with each formula on the training set and pick the conservation formula with the best loss. Formulas can be a useful, interpretable description in scientific domains given the appropriate state-space. However, for general descriptions from raw data, we would like to describe the loss with a neural network, as we describe in the next section.

**Meta-learning a neural loss function.** We propose Noether Networks, an architecture class for sequential prediction that consists of a base predictor $f_\theta : \tilde{x}_t \mapsto \tilde{x}_{t+1}$ and a meta-learned tailoring loss, parameterized as the conservation of a neural embedding $g_\phi : \tilde{x}_t \mapsto \mathbb{R}^k$. This embedding takes raw predictions as input (such as images in the case of video prediction). The conservation form induces a meta-inductive bias over potential learned tailoring losses. The Noether loss is formulated as

$$\mathcal{L}_{\text{Noether}}(x_0, \tilde{x}_{1:T}; g_\phi) = \underbrace{\sum_{t=1}^{T} |g_\phi(x_0) - g_\phi(\tilde{x}_t)|^2}_{(a)} \approx \underbrace{\sum_{t=1}^{T} |g_\phi(\tilde{x}_{t-1}) - g_\phi(\tilde{x}_t)|^2}_{(b)} \qquad (2)$$

where $x_0$ is the ground-truth input, the $\tilde{x}_t = f_\theta(\tilde{x}_{t-1})$ are the model's predictions, and $\tilde{x}_0 \triangleq x_0$. Expressions 2(a) and 2(b) are equivalent if we fully enforce the conservation law, but they differ if conservation is not fully enforced. When not fully conserving, 2(a) propagates information from ground truth more directly to the prediction, but 2(b) may be a useful approximation which better handles imperfectly conserved quantities, where the quantity should be Lipschitz but not exactly constant. In both cases, if we tailor $\theta$ with a single gradient step; the gradient update takes the form

$$\theta(x_0; \phi) = \theta - \lambda_{\text{in}} \nabla_\theta \mathcal{L}_{\text{Noether}}(x_0, \tilde{x}_{1:T}(\theta); g_\phi). \qquad (3)$$

We compute final predictions as $\hat{x}_t = f_{\theta(x_0;\phi)}(\hat{x}_{t-1})$. We can now backpropagate from $\mathcal{L}_{\text{task}}(x_{1:T}, \hat{x}_{1:T})$ back to $\phi$, which will be optimized so that the unsupervised adaptation $\theta \mapsto \theta(x_0; \phi)$ helps lower $\mathcal{L}_{\text{task}}$. The optimization requires second-order gradients to express how $\phi$ affects $\mathcal{L}_{\text{task}}$ through $\theta(x_0; \phi)$. This is similar to MAML [23], as well as works on meta-learning loss functions for few-shot learning [5] and group distribution shift [71]. Algorithm 1 provides pseudo-code.

For deep learning frameworks that allow per-example weights, such as JAX [9], the loop over sequences in Alg. 1 can be efficiently parallelized. To parallelize it for other frameworks we use the CNGRAD algorithm [4], which adapts only the Conditional Normalization (CN) layers in the inner loop. Similar to BN layers, CN only performs element-wise affine transformations: $y = x \odot \gamma + \beta$, which can be efficiently parallelized in most deep learning frameworks even with per-example $\gamma, \beta$.

Finally, even though we use the two-layer optimization typical of meta-learning, we are still in the classical single-task single-distribution setting. Noether Networks learn to impose their own inductive biases via a learned loss and to leverage it via an adaptation of its parameters $\theta$. This is useful as we often do not have access to meta-partitions that distinguish data between distributions or tasks.

**Algorithm 1** Prediction and training procedures for Noether Networks with neural conservation loss

---

**Given:** predictive model class $f$; embedding model class $g$; prediction horizon $T$; training dist. $\mathcal{D}_{\text{train}}$; batch size $N$; learning rates $\lambda_{\text{in}}, \lambda_{\text{out}}, \lambda_{\text{emb}}$; task loss $\mathcal{L}_{\text{task}}$; Noether loss $\mathcal{L}_{\text{Noether}}$

1: **procedure** PREDICTSEQUENCE$(x_0; \theta, \phi)$
2:     $\tilde{x}_0, \hat{x}_0 \leftarrow x_0, x_0$
3:     $\tilde{x}_t \leftarrow f_\theta(\tilde{x}_{t-1}) \; \forall t \in \{1, \dots, T\}$                                     ▷ Initial predictions
4:     $\theta(x_0; \phi) \leftarrow \theta - \lambda_{\text{in}} \nabla_\theta \mathcal{L}_{\text{Noether}}(x_0, \tilde{x}_{1:T}; g_\phi)$                   ▷ Inner step with Noether loss
5:     $\hat{x}_t \leftarrow f_{\theta(x_0;\phi)}(\hat{x}_{t-1}) \; \forall t \in \{1, \dots, T\}$                 ▷ Final prediction with tailored weights
6:     **return** $\hat{x}_{1:T}$

7: **procedure** TRAIN
8:     $\phi \leftarrow$ randomly initialized weights                                 ▷ Initialize weights for Noether embedding $g$
9:     $\theta \leftarrow$ randomly initialized weights                                 ▷ Initialize weights for predictive model $f$
10:    **while** not done **do**
11:        Sample batch $x_{0:T}^{(0)}, \dots, x_{0:T}^{(N)} \sim \mathcal{D}_{\text{train}}$
12:        **for** $0 \leq n \leq N$ **do**
13:            $\hat{x}_{1:T}^{(n)} \leftarrow$ PREDICTSEQUENCE$(x_0^{(n)}; \theta, \phi)$
14:        $\phi \leftarrow \phi - \lambda_{\text{emb}} \nabla_\phi \sum_{n=0}^{N} \mathcal{L}_{\text{task}}(\hat{x}_{1:T}^{(n)}, x_{1:T}^{(n)})$         ▷ Outer step with task loss for embedding
15:        $\theta \leftarrow \theta - \lambda_{\text{out}} \nabla_\theta \sum_{n=0}^{N} \mathcal{L}_{\text{task}}(\hat{x}_{1:T}^{(n)}, x_{1:T}^{(n)})$                 ▷ Outer step for predictive model
16:        **return** $\phi, \theta$

---

# 4 Experiments

Our experiments are designed to answer the following questions:

1. Can Noether Networks recover known conservation laws in scientific data?
2. Are Noether Networks useful for settings with controlled dynamics?
3. Can Noether Networks parameterize useful conserved quantities from raw pixel information?
4. How does the degree of conservation affect performance on the prediction task?

## 4.1 Experimental domains

To answer the above questions, we consider a number of different experimental domains, which we overview in this subsection, before moving on to the evaluation and results.

**Spring and pendulum from state coordinates.** Greydanus et al. [28] propose the setting of an ideal spring and ideal pendulum, which will allow us to understand the behavior of Noether Networks for scientific data where we know a useful conserved quantity: the energy. They also provide data from a real pendulum from [53]. In contrast to the ideal setups, here the data is noisy and the Hamiltonian is only approximately conserved, as the system is dissipative. For the two pendulum, input $x = (p, q) \in \mathbb{R}^2$ contains its angle $q$ and momentum $p$. Given the parameters in [28] the energy is $\mathcal{H} = 3(1 - \cos q) + p^2$, with $p^2 - 3 \cdot \cos q$ being a simpler equivalent. For the spring, input $x = (p, q) \in \mathbb{R}^2$ contains the displacement $q$ and momentum $p$ and the energy is $\mathcal{H} = \frac{1}{2}(q^2 + p^2)$. Thus, $q^2 + 1.0 \cdot p^2$ is a conserved quantity, where the coefficient $1.0$ has appropriate units.

We build on their vanilla MLP baseline and discover conservation laws that, when used for meta-tailoring, improve predictions. Since the baseline predicts $\frac{d}{dt}(x_t)$ rather than $x_{t+1}$, we apply the loss to a finite-difference approximation, i.e. $\mathcal{L}_{\text{Noether}}\left(x_t, x_t + \frac{dx}{dt}\Delta_t\right) = \mathcal{L}_{\text{Noether}}\left(x_t, x_t + f_\theta(x_t)\Delta_t\right)$.

**Domain specific language for scientific formulas.** To search over Hamiltonians, we program a simple domain specific language (DSL) for physical formulas. Since formulas in the DSL have physical meaning, each sub-formula carries its own associated physical units and is checked for validity. This allows us to significantly prune the exponential space, as done in AI-Feynman [63]. The vocabulary of the DSL is the following: `Input(i)`: returns the $i$-th input, `Operation`: one of $\{+, -, \cdot, /, \sin, \cos, x^2\}$, `Parameter(u)`: trainable scalar with units $[u]$.

Table 1: Recovering $\mathcal{H}$ for the ideal pendulum.

| Method | Description | RMSE |
|---|---|---|
| Vanilla MLP | N/A | 0.0563 |
| Noether Nets | $p^2 - 2.99\cos(q)$ | 0.0423 |
| True $\mathcal{H}$ [oracle] | $p^2 - 3.00\cos(q)$ | 0.0422 |

Table 2: Recovering $\mathcal{H}$ for the ideal spring.

| Method | Description | RMSE |
|---|---|---|
| Vanilla MLP | N/A | .0174 |
| Noether Nets | $q^2 + 1.002\,p^2$ | .0165 |
| True $\mathcal{H}$ [oracle] | $q^2 + 1.000\,p^2$ | .0166 |

Figure 2: Noether Networks can recover the energy of a real pendulum, even though it is not fully conserved. This is because they only look for quantities whose conservation helps improve predictions. Moreover, by only softly encouraging conservation, it better encodes imperfect conservations.

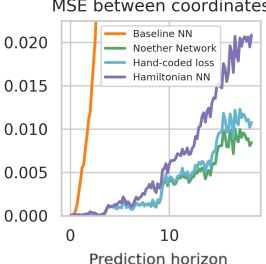
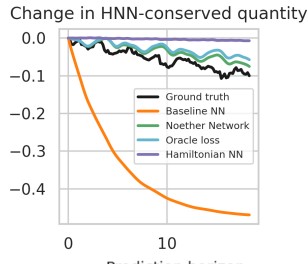

**Pixel pendulum with controls.** In settings such as robotics we may be interested in *action-conditioned* video prediction [45]: predicting future frames given previous frames and an agent's actions. We recorded videos of episodes in OpenAI Gym's pendulum swing up environment, using a scripted policy. Each model receives four history frames and a sequence of 26 policy actions starting from the current timestep, and must predict 26 future frames. As this is a visually simple environment we limit the training dataset to 5 episodes of 200 frames each, holding out 195 episodes for testing.

**Video prediction with real-world data.** To characterize the benefits provided by Noether Networks in a real-world video prediction setting, we evaluate the effect of adding a Noether conservation loss to the stochastic video generation (SVG) model [18] on the ramp scenario of the Physics 101 dataset [69]. The dataset contains videos of various real-world objects sliding down an incline and colliding with a second object. Each model is conditioned on the first two frames and must predict the subsequent 20 frames. To increase difficulty, we restrict our data to the 20-degree incline.

### 4.2 Evaluation

**Can Noether Networks recover known conservation laws in scientific data?** We first generate all valid formulas up to depth 7, removing operations between incompatible physical units and formulas equivalent up to commutativity. We obtain 41,460 and 72,372 candidate conserved formulas for the pendulum and spring, respectively, which need *not* have units of energy. These formulas are purely symbolic, with some trainable parameters still to be defined. We thus optimize their parameters via gradient descent to have low variance within sequences of true data. We then do the same for random sequences; if conservation is two orders of magnitude smaller for the true data, we accept it as an approximate conservation. This measure is similar to others used before [38], but is not sufficient when the data is noisy, the minimisation is sub-optimal, or there are numerical issues. As a result, we obtain 210 and 219 candidates for approximately conserved quantities for the pendulum and spring, respectively.

Finally, for each potentially conserved quantity, we try it as meta-tailoring loss: starting from the pre-trained vanilla MLP, we fine-tune it for 100 epochs using meta-tailoring, with one inner step and a range of inner learning rates $10^k$ for $k \in \{-3, -2.5, \ldots, 1\}$. We then evaluate the fine-tuned model on long-term predictions, keeping the expression with the best MSE loss in the training data. We observe that this process correctly singles out equations of the same form as the true Hamiltonian $\mathcal{H}$, with almost the exact parameters. Using these as losses reaches equivalent performance with that of the oracle, which uses the true formula (see Tables 1 and 2).

Finally, we run the same process for a real pendulum [53], where energy is not conserved. We use largely the same pipeline as for the ideal pendulum, the differences are explained in Appendix C. Noether Networks discover $p^2 - 2.39\cos(q)$, close to the (potentially sub-optimal) $\mathcal{H} = p^2 - 2.4\cos(q)$ described in [28]. This Noether Network improves the baseline by more than one order of magnitude, matches the performance of hand-coding the conservation loss, and improves over Hamiltonian Neural Networks, which fully impose conservation of energy. Results can be seen in figure 2.

Figure 3: In controlled pendulum environment, the Noether Network has lower mean squared error (left) and slightly better structural similarity (right). Metrics are computed and plotted by prediction timestep, where timestep 0 is a given history frame. Note that simply training SVG for more steps does not increase performance.

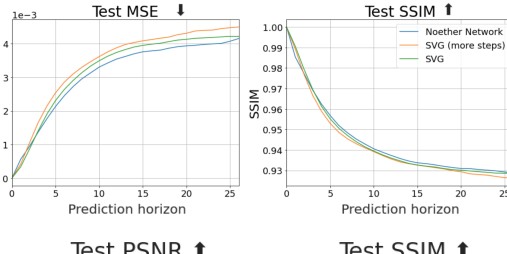

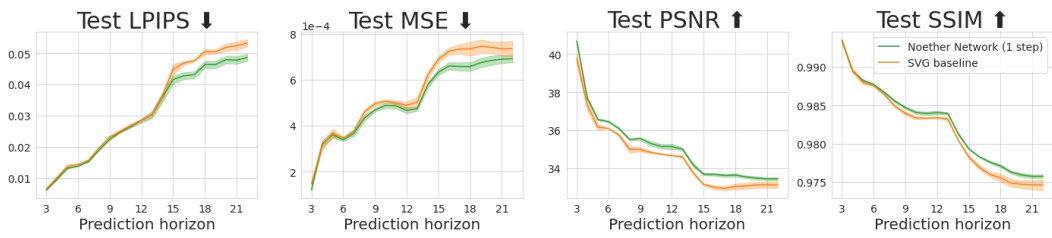

Figure 4: The Noether Network outperforms the baseline by a small margin in all four metrics in real-world video prediction, showing they can meta-learn useful conserved quantities from raw pixel data.

**Are Noether Networks useful in settings with controlled dynamics?** For the controlled pendulum experiments, we begin with an SVG model but modify it to (1) remove the prior sampling component since the environment is deterministic and (2) concatenate each action to the corresponding frame encoder output and feed the result to the LSTM. After training the SVG model for 50 epochs and fixing its weights, we run Algorithm 1 for 20 epochs to learn an embedding network for conserving useful quantities. We found that Noether Networks perform better in this setting by directly adapting the latent LSTM activations against the conservation loss, rather than adapting any network parameters. Since Noether Networks take additional training steps starting from a pre-trained SVG model, we also continued training the SVG model for 20 further epochs, which we call "SVG (more steps)." Figure 3 shows the results of each method on held out sequences. Noether Networks improve the overall mean squared error (MSE) and structural similarity (SSIM) over the base SVG.

**Can Noether Networks parameterize useful conserved quantities from raw pixel information?**
We train on only 311 training sequences, which makes generalization difficult. In this setting, the base SVG model struggles with overfitting — novel test objects often morph into objects seen at train-time, and the object in motion often morphs with the target object (as shown in figure 1). Our Noether Network uses the inner loss formulation of Equation 2(b), where $g_\phi$ is a two-layer CNN receiving two consecutive frames followed by a fully-connected projection layer producing 64-dimensional embeddings. We meta-tailor the embedding and the base model for 400 epochs. As seen in figure 4, taking a single inner tailoring step improves performance slightly over the baseline SVG model with respect to all four considered metrics: learned perceptual image patch similarity (LPIPS) [72], mean squared error (MSE), peak signal-to-noise ratio (PSNR), and structural similarity (SSIM). As the prediction horizon increases, the Noether Network performs comparatively better, likely because conservations encourage the model to retain aspects of the ground-truth input. These results provide some evidence that Noether Networks are capable of learning useful inductive biases from raw video data.

To investigate whether the Noether embeddings learn relevant features, we use Gradient-weighted Class Activation Mapping (Grad-CAM) [54] to compute importance maps for sequences in the test set. Since the Noether embedding has 64 dimensions, we perform principal component analysis (PCA) before Grad-CAM to reduce the dimensionality of the embedding, and to sort dimensions by the percentage of variance they explain in the test set frames. Interestingly, we find that the first PCA dimension captures 83.6% of the variance, and the first four dimensions capture 99.9% of the variance.

Figure 5 shows Grad-CAM localization maps for two example test-set sequences, (a) depicts an orange torus and (b) shows a blue brick, where warmer colors (red) indicate high importance and cooler colors (blue) indicate low importance. Our interpretations of Grad-CAM localization maps are consistent across examples, see Appendix F for additional ones. The first dimension, which explains the vast majority of the variance, primarily focuses on the sliding object in both examples. It also attends to the object on the table, and to the edge of the ramp. The attention to the objects suggests that the Noether Network learns to conserve quantities related to the objects' pixels and their motion (since it takes

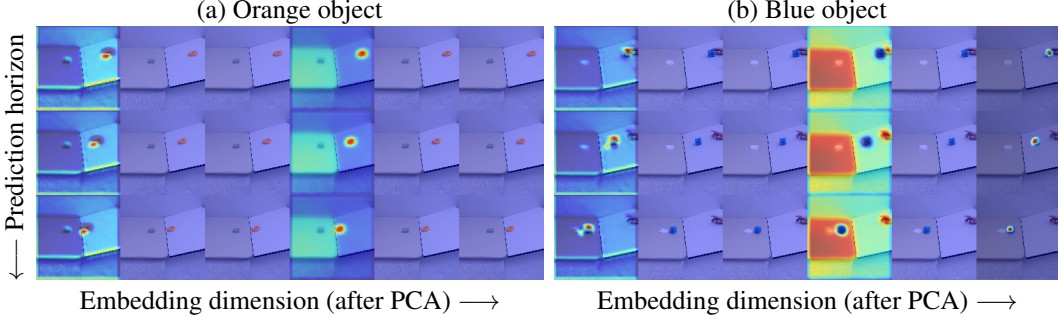

(a) Orange object          (b) Blue object

Prediction horizon

Embedding dimension (after PCA) ⟶          Embedding dimension (after PCA) ⟶

Figure 5: Grad-CAM localization maps show that the Noether embeddings attend to relevant frame regions for two example sequences. Heatmaps for the first six dimensions of the embedding (in the PCA basis, sorted by decreasing amount of variance explained) are shown for three time steps.

two consecutive frames). We hypothesize that the attention to the table edge encodes information about the orientation of the sequence, since half of the training sequences are randomly flipped horizontally. While some the dimensions focus nearly uniformly on the entire frame, the fourth dimension focuses on the orange sliding object in (a), and the human hand and table in (b). In all the test set examples, whenever the human hand is in the frame, it is attended to by this dimension. We hypothesize that it learns to conserve the hand in sequences where it is present, and possibly picks up on similarly-colored pixels as well. Finally, the sixth dimension seems to track blue sliding objects.

**How does the degree of conservation affect performance?** All of the results presented in this work were achieved by Noether Networks trained and evaluated with a single inner step. To characterize how the degree to which the conservations are imposed affects video prediction performance, we optimize the inner (Noether) loss for many inner steps and measure the outer (task) loss, as shown in figure 6. Here, the inner loss is optimized by Adam for 500 steps, as opposed to the single step of SGD during training (both settings use a learning rate of $10^{-4}$). During the optimization, the outer loss improves for roughly 150 inner steps, which suggests that the approximate conservations learned by the Noether Network generalize to more exact conservation settings.

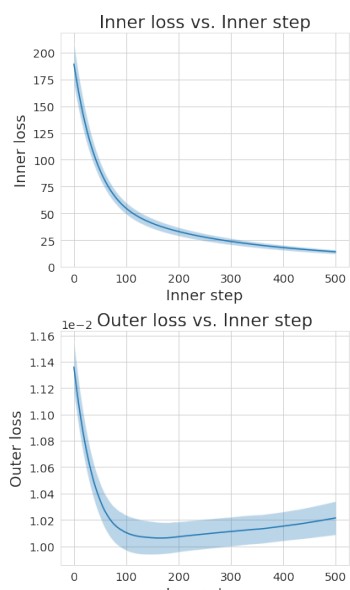

Figure 6: Despite training with only one inner step, as we lower the Noether loss to zero (top), the outer loss remains low (bottom).

## 5   Related Work

**Unsupervised adaptation and meta-learning loss functions.** Hand-designed unsupervised losses have been used to adapt to distribution shifts with rotation-prediction [59] or entropy-minimization [65], as well as to encode inductive biases [4]. Unsupervised losses have also been meta-learned for learning to encode human demonstrations [70], few-shot learning exploiting unsupervised information [41, 5], and learning to adapt to group distribution shifts [71]. In contrast, our unsupervised loss only takes the single query we care about, thus imposing no additional assumptions on top of standard prediction problems, and takes the form of a conservation law for predicted sequences.

**Encoding of symmetries and physics-based inductive biases in neural networks.** We propose to encode symmetries in dynamical systems as meta-learned conserved quantities. This falls under the umbrella of geometric deep learning, which aims to encode symmetries in neural networks; Bronstein et al. [12] provide an excellent review of the field. Most relevant to the types of problems we focus on is roto-translational equivariance [62, 66, 25, 51], applications of GNNs in physical settings [13, 6, 34, 3, 2, 49, 46, 50, 17] and the encoding of time-invariance in RNNs [22, 32, 30, 14, 60]. Recent works have encoded Hamiltonian and Lagrangian mechanics into neural models [28, 40, 15, 24], with gains in data-efficiency in physical and robotics systems, including some modeling controlled or dissipative systems [73, 19]. In contrast to these works, we propose to encode biases via prediction-time fine-tuning following the tailoring framework [4], instead of architectural constraints. This allows

us to leverage the generality of loss functions to encode a class of inductive biases extending beyond Lagrangian mechanics, and handle raw video data. In this regard, Noether networks are closer to energy-based models (EBMs) [1, 42, 29, 37, 20], which also optimize a loss function to make predictions. However, EBMs directly optimize the prediction, instead of tailoring the weights of a forward model, which can ease the learning problem. In particular, [42] show how EBMs can be a good model to enforce constraints, showing it for predicting two bouncing balls from state information.

Outside mechanics, Suh and Tedrake [58] highlight the difficulty of learning physical inductive biases with deep models from raw data and instead propose to encode mass conservation from pixels with a constrained linear model. Finally, Noether's theorem has been used [26, 61] to theoretically understand the optimization dynamics during learning; unrelated to our goal of discovering inductive biases for sequential prediction.

**Discovery of symmetries and conserved quantities.** Noether networks are related to slow feature analysis [68, 43, 7, 27, 31], which aims at extracting the smoothest features from time series in an unsupervised way, to use it for dimensionality reduction, regression, or classification. In contrast to this subfield, Noether networks impose these approximate conservations at prediction-time to improve predictions. There are other previous works which aim to learn conserved quantities in dynamical systems from data. The approach of Schmidt and Lipson [53] focuses on candidate equations that correctly predict dynamic relationships between parts of the system by measuring the agreement between the ratios of the predicted and observed partial (time) derivatives of various components of a system. A set of analytical expressions is maintained and updated as new candidates achieve sufficient accuracy. More recently, Liu and Tegmark [38] discover conservation laws of Hamiltonian systems by performing local Monte Carlo sampling followed by linear dimensionality estimation to predict the number of conserved quantities from an explained ratio diagram. Wetzel et al. [67] learn a Siamese Network that learns to differentiate between trajectories. Contrary to both, Noether Networks do not need segmentations into trajectories with different conserved quantities and can deal with raw pixel inputs and outputs.

Other approaches to learning symmetries from data include neuroevolution for learning connectivity patterns [56], regularizers for learning data augmentation [8], and meta-learning equivariance-inducing weight sharing [74]. However, these approaches do not aim to learn conserved quantities in dynamical systems. Finally, the two-step pipeline of symbolic search with unit constraints followed by gradient descent to estimate parameters has many commonalities with approaches for inductive process modeling [36, 11, 10] for scientific discovery. However, Noether networks then leverage the discovered physical equations to improve the predictions of a neural network.

**Neural networks to discover physical laws.** There has been a growing interest in leveraging the functional search power of neural networks for better scientific models. Multiple works train a neural network and then analyze it to get information, such as inferring missing objects via back-propagation descent [3], inspecting its gradients to guide a program synthesis approach [63], learning a blue-print with a GNN [16], or leveraging overparametrized-but-regularized auto-encoders [64]. Others, such as DreamCoder [21], take the explicit approach with a neural guided synthesis over the space of formulas. Unlike these works, our method can readily be applied both to symbolic conservation laws and to neural network conservation losses applied to raw videos.

## 6    Discussion

**Limitations.** A disadvantage of our proposed approach over architectural constraints (in the cases when the latter can represent the same bias) is computation time. Noether Networks optimize the conservation loss inside the prediction function, which results in an overhead. This can be reduced by only fine-tuning the higher-level layers, avoiding back-propagation back to the input. Another approach to reduce training time could be to warm-start the Noether embeddings with pre-trained contrastive embeddings, which have an aligned goal and have shown great results for video and audio classification [48]. It is also sometimes the case that, by encoding the inductive bias during training the trained but un-tailored Noether Network performs better than a comparable architecture without the bias during training. It is also worth noting that the theoretical guarantees apply to fully enforced conservation constraints, while in practice we only fine-tune our model to better satisfy them. However, we believe the theoretical result still conveys the potential benefits of Noether networks.

**Broader impact.** We propose a new framework for meta-learning inductive biases in sequential settings, including from raw data. On the positive side, our framework can be used to discover conserved quantities in scientific applications, even when these are not fully conserved. By improving video prediction, we are also empowering predictive models for entities such as robots and self-driving cars, helping elder care and mobility. However, at the same time, it could be used for better predicting the movement and tracking of people from CCTV cameras, affecting privacy. It could also be used to improve the quality of deepfake video generation, which can be used for malicious purposes.

**Overview and future work.** We propose Noether Networks: a new framework for meta-learning inductive biases through conserved quantities imposed at prediction time. Originally motivated by applications in physics, our results indicate that Noether networks can recover known conservation laws in scientific data and also provide modest gains when applied to video prediction from raw pixel data. The generality of optimizing arbitrary unsupervised losses at prediction time provides an initial step towards broader applications. Finally, this work points at the usefulness of designing meta-learned inductive biases by putting priors on the biases instead of hard-coding biases directly.

## Acknowledgements

We want to thank Leslie Pack Kaelbling, Tomás Lozano-Pérez and Maria Bauza for their useful and detailed feedback, as well as Yannic Kilcher, Max Welling, and Joel Zylberberg for insightful comments. We gratefully acknowledge support from GoodAI; from NSF grant 1723381; from AFOSR grant FA9550-17-1-0165; from ONR grants N00014-18-1-2847 and N00014-21-1-2685; from the Honda Research Institute, from MIT-IBM Watson Lab; from SUTD Temasek Laboratories; and from Google. Chelsea Finn is a fellow in the CIFAR Learning in Machines and Brains program. We also acknowledge the MIT SuperCloud and Lincoln Laboratory Supercomputing Center for providing HPC resources that have contributed to the reported research results. Any opinions, findings, and conclusions or recommendations expressed in this material are those of the authors and do not necessarily reflect the views of our sponsors.

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
