# A Proofs

We utilize the following lemma in the proof of Theorem 1.

**Lemma 1.** *Fix $v \in \mathbb{R}^d$. Then, for any $\delta > 0$, with probability at least $1 - \delta$ over an i.i.d. draw of $n$ examples $((x_i, y_i))_{i=1}^n$, the following holds:*

$$\mathbb{E}_{x,y}[\mathcal{L}(f(x, v), y)] - \frac{1}{n} \sum_{i=1}^n \mathcal{L}(f(x_i, v), y_i) \leq C \sqrt{\frac{\ln(1/\delta)}{2n}}.$$

*Proof of Lemma 1.* By using Hoeffding's inequality, we have that

$$\Pr\left( \mathbb{E}_{x,y}[\mathcal{L}(f(x), y)] - \frac{1}{n} \sum_{i=1}^n \mathcal{L}(f(x_i, v), y_i) \geq t \right) \leq \exp\left( -\frac{2nt^2}{C^2} \right),$$

where $t \geq 0$. Solving $\delta = \exp\left( -\frac{2nt^2}{C^2} \right)$ for $t \geq 0$, we have that for any $\delta > 0$, with probability at least $1 - \delta$, the following holds:

$$\mathbb{E}_{x,y}[\mathcal{L}(f(x), y)] - \frac{1}{n} \sum_{i=1}^n \mathcal{L}(f(x_i, v), y_i) \leq C \sqrt{\frac{\ln(1/\delta)}{2n}}.$$

$\square$

## A.1 Proof of Theorem 1

### A.1.1 Preparation

In this subsection, we focus on the case of $g_\phi(f_\theta(x)) \neq g_\phi(x)$ as a preparation for the more general case in the next subsection. The (closed) ball of radius $r$ centered at $c$ is denoted by $\mathcal{B}_r[c] = \{v \in \mathbb{R}^d : \|v - c\|_2 \leq r\}$. Fix $r > 0$ and $\mathcal{C}(r, \mathcal{V}) \in \arg\min_{\mathcal{C}}\{|\mathcal{C}| : \mathcal{C} \subseteq \mathbb{R}^d, \mathcal{V} \subseteq \cup_{c \in \mathcal{C}} \mathcal{B}_r[c]\}$. Let $\mathcal{N}(r, \mathcal{V}) = |\mathcal{C}(r, \mathcal{V})|$; i.e. the minimum number of balls of radius $r$ needed to cover the a set of vectors $\mathcal{V}$.

The statement of theorem 1 vacuously holds if $R$ is unbounded. Thus, we focus on the case of $R < \infty$ in the rest of the proof. For any $\theta \in \Theta$, there exists $v \in \mathcal{V}$ such that

$$\mathbb{E}_{x,y}[\mathcal{L}(f_\theta(x), y)] - \frac{1}{n} \sum_{i=1}^n \mathcal{L}(f_\theta(x_i), y_i) = \mathbb{E}_{x,y}[\mathcal{L}(f(x, v), y)] - \frac{1}{n} \sum_{i=1}^n \mathcal{L}(f(x_i, v), y_i) = \psi(v). \tag{4}$$

Moreover, for any $v \in \mathcal{V}$, the following holds: for any $c \in \mathcal{C}(r, \mathcal{V})$,

$$\psi(v) = \psi(c) + (\psi(v) - \psi(c)). \tag{5}$$

For the first term in the right-hand side of (5), by using Lemma 1 with $\delta \to \delta/\mathcal{N}(r, \mathcal{V})$ and taking union bounds, we have that for any $\delta > 0$, with probability at least $1 - \delta$, the following holds for all $c \in \mathcal{C}(r, \mathcal{V})$:

$$\psi(c) \leq C \sqrt{\frac{\ln(\mathcal{N}(r, \mathcal{V})/\delta)}{2n}}. \tag{6}$$

By combining equations (5) and (6), we have that for any $\delta > 0$, with probability at least $1 - \delta$, the following holds for any $v \in \mathcal{V}$ and all $c \in \mathcal{C}(r, \mathcal{V})$:

$$\psi(v) \leq C \sqrt{\frac{\ln(\mathcal{N}(r, \mathcal{V})/\delta)}{2n}} + (\psi(v) - \psi(c)). \tag{7}$$

This implies that for any $\delta > 0$, with probability at least $1 - \delta$, the following holds for any $v \in \mathcal{V}$:

$$\psi(v) \leq C \sqrt{\frac{\ln(\mathcal{N}(r, \mathcal{V})/\delta)}{2n}} + \min_{c \in \mathcal{C}(r, \mathcal{V})} |\psi(v) - \psi(c)|. \tag{8}$$

For the second term in the right-hand side of (8), we have that for any $v \in \mathcal{V}$,

$$\min_{c \in \mathcal{C}(r, \mathcal{V})} |\psi(v) - \psi(c)| \leq \zeta \min_{c \in \mathcal{C}(r, \mathcal{V})} \|v - c\|_2 \leq \zeta r. \tag{9}$$

Thus, by using $r = \zeta^{1/\rho - 1} \sqrt{\frac{1}{n}}$, we have that for any $\delta > 0$, with probability at least $1 - \delta$, the following holds for all $v \in \mathcal{V}$:

$$\psi(v) \leq C \sqrt{\frac{\ln(\mathcal{N}(r, \mathcal{V})/\delta)}{2n}} + \sqrt{\frac{\zeta^{2/\rho}}{n}}. \tag{10}$$

Using equation (4), this implies that for any $\delta > 0$, with probability at least $1 - \delta$, the following holds for all $\theta \in \Theta$:

$$\mathbb{E}_{x,y}[\mathcal{L}(f_\theta(x), y)] - \frac{1}{n} \sum_{i=1}^{n} \mathcal{L}(f_\theta(x_i), y_i) \leq C \sqrt{\frac{\ln(\mathcal{N}(r, \mathcal{V})/\delta)}{2n}} + \sqrt{\frac{\zeta^{2/\rho}}{n}}, \tag{11}$$

where $r = \zeta^{1/\rho - 1} \sqrt{\frac{1}{n}}$. Since $\mathcal{N}(r, \mathcal{V}) \leq (2R\sqrt{d}/r)^d = (2R(\zeta^{1-1/\rho})\sqrt{nd})^d$,

$$\mathbb{E}_{x,y}[\mathcal{L}(f_\theta(x), y)] - \frac{1}{n} \sum_{i=1}^{n} \mathcal{L}(f_\theta(x_i), y_i)$$

$$\leq C \sqrt{\frac{\ln(\mathcal{N}(r, \mathcal{V})/\delta)}{2n}} + \sqrt{\frac{\zeta^{2/\rho}}{n}}$$

$$= C \sqrt{\frac{\ln(\mathcal{N}(r, \mathcal{V})) + \ln(1/\delta)}{2n}} + \sqrt{\frac{\zeta^{2/\rho}}{n}},$$

$$= C \sqrt{\frac{\ln((2R(\zeta^{1-1/\rho})\sqrt{nd})^d) + \ln(1/\delta)}{2n}} + \sqrt{\frac{\zeta^{2/\rho}}{n}},$$

$$= C \sqrt{\frac{d\ln(\sqrt{d}) + d\ln(2R(\zeta^{1-1/\rho})\sqrt{n}) + \ln(1/\delta)}{2n}} + \sqrt{\frac{\zeta^{2/\rho}}{n}}.$$

### A.1.2 Putting results together

In this subsection, we now generalize the proof of the previous subsection to the case of $g_\phi(f_\theta(x)) = g_\phi(x)$. The condition of $g_\phi(f_\theta(x)) = g_\phi(x)$ implies that

$$g_\phi^{-1}[\{g_\phi(f_\theta(x))\}] = g_\phi^{-1}[\{g_\phi(x)\}]. \tag{12}$$

Since $g_\phi^{-1}[\{g_\phi(x)\}] = \{x + Az : z \in \mathbb{R}^m\}$ and $f_\theta(x) = x + v_\theta$, the right-hand side of equation (12) can be simplified as

$$g_\phi^{-1}[\{g_\phi(f_\theta(x))\}] = \{f_\theta(x) + Az : z \in \mathbb{R}^m\} = \{x + v_\theta + Az : z \in \mathbb{R}^m\}.$$

Substituting this into equation (12) yields

$$\{x + v_\theta + Az : z \in \mathbb{R}^m\} = g_\phi^{-1}[\{g_\phi(x)\}] = \{x + Az : z \in \mathbb{R}^m\}, \tag{13}$$

where the last equality uses the fact that $g_\phi^{-1}[\{g_\phi(x)\}] = \{x + Az : z \in \mathbb{R}^m\}$. Equation (13) implies that

$$v_\theta \in \mathrm{Col}(A) \subseteq \mathbb{R}^d, \tag{14}$$

where $\mathrm{Col}(A)$ is the column space of the matrix $A \in \mathbb{R}^{d \times m}$. Let $\bar{A} \in \mathbb{R}^{d \times m}$ be a semi-orthogonal matrix such that $\bar{A}^\top \bar{A} = I_m$ (i.e., the identity matrix of size $m$ by $m$) and $\mathrm{Col}(\bar{A}) = \mathrm{Col}(A)$. Then, equation (14) implies that for any $\theta \in \Theta$, there exists $z \in \mathbb{R}^m$ such that

$$v_\theta = \bar{A}z. \tag{15}$$

We can further refine this statement by using the following observation. Since $\bar{A}^\top \bar{A} = I_m$, we have that

$$R \geq \|v_\theta\|_2 = \|\bar{A}z\|_2 = \|z\|_2, \tag{16}$$

and

$$|\psi(\bar{A}z) - \psi(\bar{A}z')| \leq \zeta\|\bar{A}(z - z')\|_2 = \zeta\|z - z'\|_2. \tag{17}$$

Define $\mathcal{Z} = \{z \in \mathbb{R}^m : \|z\|_2 \leq R\}$. Then, equations (14) and (16) together imply that for any $\theta \in \Theta$, there exists $z \in \mathcal{Z}$ such that

$$v_\theta = \bar{A}z. \tag{18}$$

Whereas the previous subsection defines the ball in the space of $v \in \mathbb{R}^d$, we can now define the ball in the space of $z \in \mathbb{R}^m$ to cover the space of $v \in \mathbb{R}^d$ through equation (18). That is, we re-define $\mathcal{B}_r[c] = \{z \in \mathbb{R}^m : \|z - c\|_2 \leq r\}$. Fix $r > 0$ and $\mathcal{C}(r, \mathcal{Z}) \in \operatorname{argmin}_{\mathcal{C}}\{|\mathcal{C}| : \mathcal{C} \subseteq \mathbb{R}^m, \mathcal{Z} \subseteq \cup_{c \in \mathcal{C}}\mathcal{B}_r[c]\}$. Let $\mathcal{N}(r, \mathcal{Z}) = |\mathcal{C}(r, \mathcal{Z})|$. Using (18), for any $\theta \in \Theta$, there exists $z \in \mathcal{Z}$ such that

$$\mathbb{E}_{x,y}[\mathcal{L}(f_\theta(x), y)] - \frac{1}{n}\sum_{i=1}^{n}\mathcal{L}(f_\theta(x_i), y_i) = \mathbb{E}_{x,y}[\mathcal{L}(f(x, \bar{A}z), y)] - \frac{1}{n}\sum_{i=1}^{n}\mathcal{L}(f(x_i, \bar{A}z), y_i)$$
$$= \psi(\bar{A}z), \tag{19}$$

Moreover, for any $z \in \mathcal{Z}$, the following holds: for any $c \in \mathcal{C}(r, \mathcal{Z})$,

$$\psi(\bar{A}z) = \psi(\bar{A}c) + (\psi(\bar{A}z) - \psi(\bar{A}c)). \tag{20}$$

For the first term in the right-hand side of (20), by using Lemma 1 with $\delta \to \delta/\mathcal{N}(r, \mathcal{Z})$ and taking union bounds, we have that for any $\delta > 0$, with probability at least $1 - \delta$, the following holds for all $c \in \mathcal{C}(r, \mathcal{Z})$:

$$\psi(\bar{A}c) \leq C\sqrt{\frac{\ln(\mathcal{N}(r, \mathcal{Z})/\delta)}{2n}}. \tag{21}$$

By combining equations (20) and (21), we have that for any $\delta > 0$, with probability at least $1 - \delta$, the following holds for any $z \in \mathcal{Z}$ and all $c \in \mathcal{C}(r, \mathcal{Z})$:

$$\psi(\bar{A}z) \leq C\sqrt{\frac{\ln(\mathcal{N}(r, \mathcal{Z})/\delta)}{2n}} + (\psi(\bar{A}z) - \psi(\bar{A}c)). \tag{22}$$

This implies that for any $\delta > 0$, with probability at least $1 - \delta$, the following holds for any $z \in \mathcal{Z}$:

$$\psi(\bar{A}z) \leq C\sqrt{\frac{\ln(\mathcal{N}(r, \mathcal{Z})/\delta)}{2n}} + \min_{c \in \mathcal{C}(r, \mathcal{Z})}|\psi(\bar{A}z) - \psi(\bar{A}c)|. \tag{23}$$

For the second term in the right-hand side of (23), by using equation (17), we have that for any $z \in \mathcal{Z}$,

$$\min_{c \in \mathcal{C}(r, \mathcal{Z})}|\psi(\bar{A}z) - \psi(\bar{A}c)| \leq \zeta \min_{c \in \mathcal{C}(r, \mathcal{V})}\|z - c\|_2 \leq \zeta r.$$

If $m = 0$, then since $\mathcal{N}(r, \mathcal{Z}) = 1$ with $r = 0$,

$$\mathbb{E}_{x,y}[\mathcal{L}(f_\theta(x), y)] - \frac{1}{n}\sum_{i=1}^{n}\mathcal{L}(f_\theta(x_i), y_i) \leq C\sqrt{\frac{\ln(1/\delta)}{2n}}.$$

This proves the desired statement for $m = 0$. Thus, we focus on the case of $m \geq 1$ in the rest of the proof. By using $r = \zeta^{1/\rho - 1}\sqrt{\frac{1}{n}}$, we have that for any $\delta > 0$, with probability at least $1 - \delta$, the following holds for all $z \in \mathcal{Z}$:

$$\psi(\bar{A}z) \leq C\sqrt{\frac{\ln(\mathcal{N}(r, \mathcal{Z})/\delta)}{2n}} + \sqrt{\frac{\zeta^{2/\rho}}{n}}. \tag{24}$$

Using equation (19), this implies that for any $\delta > 0$, with probability at least $1 - \delta$, the following holds for all $\theta \in \Theta$:

$$\mathbb{E}_{x,y}[\mathcal{L}(f_\theta(x), y)] - \frac{1}{n}\sum_{i=1}^{n}\mathcal{L}(f_\theta(x_i), y_i) \leq C\sqrt{\frac{\ln(\mathcal{N}(r, \mathcal{Z})/\delta)}{2n}} + \sqrt{\frac{\zeta^{2/\rho}}{n}}, \tag{25}$$

where $r = \zeta^{1/\rho - 1}\sqrt{\frac{1}{n}}$. Since $\mathcal{N}(r, \mathcal{Z}) \leq (2R\sqrt{m}/r)^m = (2R(\zeta^{1-1/\rho})\sqrt{nm})^m$,

$$\mathbb{E}_{x,y}[\mathcal{L}(f_\theta(x), y)] - \frac{1}{n}\sum_{i=1}^{n}\mathcal{L}(f_\theta(x_i), y_i)$$

$$\leq C\sqrt{\frac{\ln(\mathcal{N}(r, \mathcal{Z})/\delta)}{2n}} + \sqrt{\frac{\zeta^{2/\rho}}{n}}$$

$$= C\sqrt{\frac{\ln(\mathcal{N}(r, \mathcal{Z})) + \ln(1/\delta)}{2n}} + \sqrt{\frac{\zeta^{2/\rho}}{n}},$$

$$= C\sqrt{\frac{\ln((2R(\zeta^{1-1/\rho})\sqrt{nm})^m) + \ln(1/\delta)}{2n}} + \sqrt{\frac{\zeta^{2/\rho}}{n}},$$

$$= C\sqrt{\frac{m\ln(\sqrt{m}) + m\ln(2R(\zeta^{1-1/\rho})\sqrt{n}) + \ln(1/\delta)}{2n}} + \sqrt{\frac{\zeta^{2/\rho}}{n}}.$$

$\square$

# B  Why Noether Networks avoid learning trivial conserved quantities

Neural networks have large capacity. Therefore, we have to be careful that the Noether embedding is not learning a quantity that is trivially conserved but is unrelated to the task, such as predicting a constant vector $g_\phi(x) = \vec{C}$. In our setup both the prediction network and the meta-learned conservation embedding are trained end-to-end to minimize the task loss after the conservation update. However, for the meta-learned embedding this is equivalent to encouraging a big decrease in supervised loss after the update than before the update. This can be seen by looking at the gradient of the difference between the task loss before and after the update:

$$\nabla_\phi \left( \mathcal{L}_{\text{task}}(x_{1:T}, f_{\theta(x_0;\phi)}) - \mathcal{L}_{\text{task}}(x_{1:T}, f_\theta) \right)$$

Here $\theta(x_0; \phi)$ denotes the model parameters after the inner loss update which conserves the embedding $g_\phi$ for input $x_0$. Notice that the second loss does not depend on the embedding (since $\theta$ has not yet been tailored with $g_\phi$). Thus, the gradient of the overall expression is equivalent to:

$$\nabla_\phi \left( \mathcal{L}_{\text{task}}(x_{1:T}, f_{\theta(x_0;\phi)}) \right)$$

which is the one we optimize. A trivially conserved quantity (like $g_\phi(x) = 0$ independent of $\phi$ and $x$) will not produce any improvement on the supervised loss after its conservation is encouraged. Therefore, the embedding will seek to be more useful, by making the tailored weights $\theta(x_0; \phi)$ perform better than the untailored ones $\theta$.

# C  Experimental details for scientific data

We use three different experiments from [28]: the ideal pendulum, the ideal spring, and the real pendulum, under an Apache license. The first two come from simulated ODEs without noise, the latter comes from a real data from Schmidt and Lipson [53]. It is worth noting that other datasets from Greydanus et al. [28] (such as a planetary system) could not be included because the DSL required too much depth to reach the conserved energy quantity. A better search, such as using evolution, or a better DSL (such as those derived from many scientific formulas in DreamCoder [21]) could remedy this. Finally, note that an approach that searched over the same DSL encoding the loss as a generic $f(x_0, x_t)$ loss, not as a conservation $|g(x_0) - g(x_t)|$ would require more than twice the depth, thus not being able to cover the evaluated datasets.

Most experimental details and hyperparameters are already detailed in the main text. It is worth noting that the pipeline approach: first discovering approximately conserved quantities (which had at least 2 orders of magnitude less variance in real vs. random data), then picking the best loss when used with meta-tailoring. This choice of 2 orders of magnitude comparing variance in real data vs. random data, was heuristically chosen to speed up the search, without trying other hyper-parameters.

The experiments with the real pendulum differ slightly from the experiments with the ideal pendulum due to differences in the data: the real pendulum training sequences are continuous, and they are 4

times as long as the test sequences. Therefore, we divide the training sequences into 4 equally-long trajectories and choose the configuration with the best long-term (not single-step) *training* loss. Second, because the data is noisy and we are only trying to enforce approximate conservations, we tried different inner learning rates $(10^{-3}, 10^{-2}, 10^{-1}, 1)$ as well as number of inner steps $(1, 2, 3)$.

Experiments were performed on an NVIDIA Tesla V-100 GPU and 10 CPU cores, taking around 4 hours to run.

**Extra details on state space and energies** For the pendulum, input $x = (p, q) \in \mathbb{R}^2$ contains its angle $q$ and momentum $p$. The formula for the energy is $\mathcal{H} = 2mgl(1 - \cos q) + \frac{l^2 p^2}{2m}$. Greydanus et al. [28] set $m = \frac{1}{2}, l = 1, g = 3$, resulting in $\mathcal{H} = 3(1 - \cos q) + p^2$. Notice that a simpler conserved quantity is $p^2 - 3 \cdot \cos q$. For the spring, input $x = (p, q) \in \mathbb{R}^2$ contains the displacement $q$ and momentum $p$, and the system's energy is given by $\mathcal{H} = \frac{1}{2}kq^2 + \frac{p^2}{2m}$. Greydanus et al. [28] set $k = q = m = 1$, resulting in $\mathcal{H} = \frac{1}{2}(q^2 + p^2)$, where units are omitted. Thus, $q^2 + 1 \cdot p^2$ is a conserved quantity, where coefficient 1 has appropriate units.

For the real pendulum, it is worth noting that the energy candidate $p^2 - 2.4 \cos q$ was proposed by humans to fit the data, since the data is experimental. It could thus be slightly sub-optimal, which could explain why Noether Networks improves its predictions. We also consider the possibility of it being due to fluctuations in noisy data and it being a small dataset.

## D    Experimental Details for pendulum with controls

We generated the action-conditioned video prediction data in OpenAI Gym's "Pendulum-v0" environment. We want to make each episode different and not easily predictable, but also not too erratic. Hence we generate actions for each episode by:

$$a_t = 2 \sin(\omega t + \alpha)$$

where $\omega, \alpha$ are randomly sampled for each episode, with $\omega \sim \mathrm{U}[0.05, 0.15]$ and $\alpha \sim \mathrm{U}[0, 2\pi]$. We recorded 200 episodes of 200 timesteps each and split them into 5 train and 195 test videos.

Our action conditioned video prediction architecture is based off of the publicly available implementation[2] of SVG, with prior sampling removed and actions concatenated to the output of the frame encoder. The frame encoder and decoder are based on the DCGAN generator and discriminator, and the latent frame predictor is a 2-layer LSTM with hidden size 256. We resize the input frames to $64 \times 64$ before inputting to the frame encoder. Throughout all experiments we use a batch size of 32 and optimize using Adam with a learning rate of 0.001.

During evaluation, each method receives a length-4 frame history and 26 future actions and must predict the next 26 frames. However, during training each method only predicts the next 10 frames and then is updated to minimize mean square error; this speeds up each step and reduces memory cost. For the Noether Network there is an inner loop where we minimize the learned conservation loss: in these experiments, we found it better to update the LSTM's initial outputs during the inner loop, rather than any network parameters. We then pass the updated LSTM outputs into the frame decoder to generate a video prediction. The inner loop optimization is done with gradient descent using a learning rate of 0.5, for 1 step. As described above the conservation loss involves conserving a learned embedding over time. Our learned embedding in this case is a 3-layer MLP with 512 units at each hidden layer and outputs an embedding with dimension 100. The embedding receives as input the LSTM output for the past two time steps and the most recent action, all concatenated together.

We ran all the action conditioned experiments on a single machine with an NVIDIA RTX 2080 Ti GPU. Due to the limited data available, all training was relatively fast: the initial 50 epochs of SVG training took $\sim 40$ minutes, while the Noether Network tuning process took $\sim 17$ minutes.

## E    Experimental Details for Physics 101

We use a subset of the Physics 101 dataset [69] with videos from the ramp scenario, where various objects are let go by a human hand at the top of a ramp. We could not find its associated license. There

---

[2]https://github.com/edenton/svg

are two ramp settings: 10 degrees and 20 degrees. To ensure the prediction problem is interesting and nontrivial, we only take examples with the 20 degree ramp since the object does not slide down the ramp in many of the 10 degree examples. Additionally, we only take sequences with length at least 2 seconds, and use a frame rate of 15 frames per second when extracting frames to pass to the models. We use videos recorded with the `Kinect_RGB_1` sensor. Video frames are center-cropped at full height (1080 pixels) and downsample to obtain $128 \times 128$ images, and perform random horizontal flipping for sequences. This preprocessing results in 389 possible sequences, and we take a random 80/20 train/test split. In the prediction task, we condition the model on two frames, and the model must predict the subsequent 20 frames. To ensure that we extract segments of videos where the object is moving, we take the middle 22 frames of each video clip. We use a mini-batch size of 2 for these experiments.

Our baseline model is based on the publicly-available implementation of SVG-LP [18]. As in the original implementation, the encoder and decoder are based on VGG16 [55], the latent frame predictor is a 2-layer LSTM with hidden size of 128, and both the prior and posterior are single-layer LSTMs that produce latent variables $\mathbf{z}_t$ of dimension 64. We replace all batch normalization layers with layer normalization layers in both the baseline and the Noether Network because of the small mini-batch size used in our experiments. Preliminary experiments showed similar performance, but much more stable training. The learning rate used during the training of the baseline 0.0001. Teacher forcing is used during the training of the SVG baseline and the meta-training of the Noether Network, as done by Denton and Fergus [18].

The meta-learned inner loss of the Noether Network is formulated as in equation 2(b), with the learned embedding $g_\phi$ parameterized as a 2-layer convolutional network where each layer consists of a $5 \times 5$ convolution with 32 filters in the first layer and 64 filters in the second layer, a ReLU non-linearity, and $2 \times 2$ max pooling. There is a final linear layer which projects onto a 64-dimensional embedding space.

The CNGRAD algorithm of Alet et al. [4] is used to perform tailoring. In the Noether Network, conditional normalization (CN) layers are inserted after each batch normalization layer in both the encoder and the decoder. The CN parameters are initialized to the identity transformation, and are adapted with a single inner step at prediction-time using an inner learning rate of 0.0001. The embedding is trained in the outer loop with an outer learning rate of 0.0001. We report results obtained with meta-tailoring, where both the base model and the embedding are randomly initialized.

We tried multiple Noether embedding dimensions as well as inner learning rates, but results were very consistent across these choices so we left our original choices. The only sets which failed to learn were very low inner learning rates ($\leq 10^{-5}$) with very high ($> 1024$) embedding dimensions. We conjecture this stability is due to two reasons:

1. Meta-learning the loss function compensates for the choice of inner learning rate: for instance we observed that increasing the inner learning rate by two orders of magnitude resulted in two orders of magnitude smaller losses (and thus equal gradient updates).

2. PCA analysis of the Noether embeddings suggests they fundamentally encoded few ($\approx 4$) dimensions, even when the embeddings had more dimensions. More work needs to be done to understand whether this is a fundamental property of the domain discovered by the method or an optimization issue.

We ran all experiments with the Physics 101 dataset on an NVIDIA Tesla V100 GPU and 20 CPU cores. Training the baseline until convergence took 400 epochs and approximately 13 hours, and training the Noether Network took approximately 2 days, 23 hours. The error bars reported in figure 4 and figure 6 are standard error of the mean (SEM) for $N = 5$ training seeds.

## F  Grad-CAM localization maps

In figure 7, Grad-CAM heatmaps are shown for a single time step from each of five example sequences from the test set. The trends discussed in section 4.2 are consistent across these examples as well.

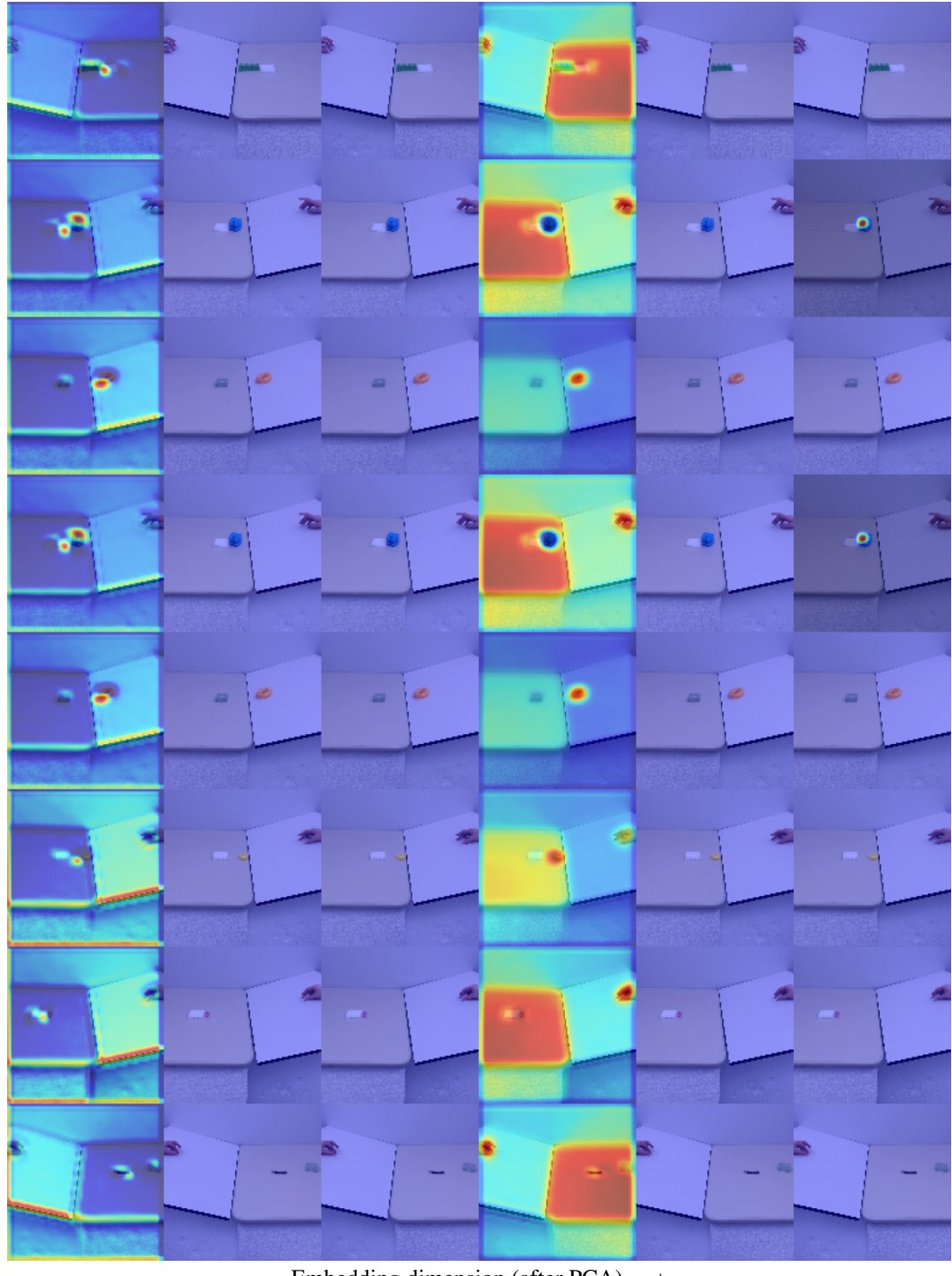

Embedding dimension (after PCA) ⟶

Figure 7: Grad-CAM localization maps have consistent behavior across sequences in the test set. Each row corresponds to a different sequence, and each column represents an embedding dimension (first PCA dimension on the left).