# OpenReview forum: "Noether Networks: meta-learning useful conserved quantities"
_NeurIPS.cc/2021/Conference — NeurIPS 2021 Poster_

### Official Review · Reviewer_HJHv · 2021-07-12

**Rating:** 5
**Confidence:** 3

**Summary:**

The authors tackle the challenge of automatically discovering symmetries in dynamical systems. As continuous symmetries are equivalently described by conserved quantities--as stated by Noether's theorem--they introduce a method for learning (approximately) conserved quantities. They do so by building on the `meta-tailoring' framework which allows to fine-tune the model according to the learned conservation laws at test time. For problems with few and interpretable coordinates (e.g. position, momentum, etc) they propose a symbolic approach that generate many valid (i.e. with compatible physical units) expressions that are parametrized with learnable weights. For raw-pixel data, they parametrize and learn the function to be conserved with a neural network. In both cases, the learned conservation function is used to guide the model towards physically 'realistic' solutions. They empirically assess their method on pendulum and spring systems, but also on real data from the Physics 101 dataset.

**Limitations And Societal Impact:**

Limitations are discussed.

**Main Review:**


## Strengths
The proposed algorithm is of relevance to the community as incorporating physical inductive bias by hand is not always possible. Empirical assessment shows that the model is able to recover the true conserved quantity and that generalization performance is indeed improved. Authors also show that enforcing a `soft conservation' is useful as dissipative systems would only approximately conserve energy.
The theoretical evidence from Section 2 strengthen the motivation of the proposed approach.

## Weaknesses
The two main weaknesses that I can identify are;
1/ clarity, as developed later.
2/ the symbolic approach to learn the conserved expression. It is an interesting approach and it seems to work well in practice, but it has a bit of a heuristic flavour and has a non marginal computational overhead. Also, I am not entirely convinced that there exist problems of interest where there is a closed-form conserved quantity that we wouldn't know about. I may be wrong though.
Also, I am not really convinced by the Physics 101 experiment as the improvement seems marginal (hard to tell without confidence intervals) and there is no visualisation of the dataset, predictions or learned embeddings.

## Clarity
Generally, I feel that even though the paper is fairly well written some of the core ideas could and should be better conveyed. For instance, readers that aren't familiar with `meta-tailoring may struggle at Section 3.1. Since this is a core aspect of the proposed method, it would be worth adding an equation or a diagram guiding the reader.
Also, regarding Section 2, the first paragraph is quite packed with notations. I am not certain that it is necessary to introduce general notations as a special case is eventually considered. Also, a simple diagram could be useful to illustrate this setting and give intuition to the reader on the improved generalization gap. If I understand correctly, the main idea of this section is that, if the solution space is an affine subspace, and this is enforced, the the generalization gap bound grows w.r.t. the dimension of the subspace instead of the embedding space. I think that if this is indeed the core message, it should be stressed more.
Algorithm 1 is extremely welcomed as it is really helpful to understand the proposed approach.

## Reproducibility
No code has been provided in the supplementary materials, yet there seems to be quite some experimental details in Section 4. I am not certain that this is enough to reproduce the results.

## Additional feedback
- Links are broken.
- Lack of standard errors / confidence intervals.
- 53: "are"?
- Section 2: Is this setting motivated by a physical problem? Or because it is simple enough so that such analysis can be done.
- 65-66: What is the difference between $\phi$ and $G$?
- Equations 1 and 2: It should be stressed that the key difference is that $d$ is replaced by $m$, which is great if $m \le d$ (as stated on line 67).
- 76: It may not be immediate for the reader to understand what are the dimensions $d$ and $m$. A drawing may help?
- Figure 2 & Figure 3: Looks a bit blurry and the font is quite small.
- 246-161: It would be quite helpful to see what the 'Physics 101 ramp data' look like. Would there be a way also to visualize the learnt embeddings? Am whether it is correlated to the `true' conserved quantities (i.e. energy) in this setting?


**Time Spent Reviewing:**

4

---

> ### Author Response · Authors · 2021-08-11
> **Response to Reviewer HJHv**
>
> Thank you for your detailed review and constructive comments.
>
> **Reducing decisions for symbolic experiments** As you say, the symbolic approach is computationally expensive (the other experiments have a much more efficient gradient-based meta-learning). This forced us to take a pipeline approach to avoid training tens of thousands of tasks by first filtering approximately conserved quantities, choosing those for which tailoring worked best, and finally finding the one loss that worked best when used in meta-tailoring. To simplify the approach we have removed one of the steps and have tried all approximately conserved quantities in meta-tailoring. After filtering the conservation losses that made the ODE simulation diverge and adding more compute, we obtained the same results. This is not surprising given that if a loss performed poorly in tailoring, it also performed poorly in meta-tailoring, thus not outperforming our previously discovered loss.
>
> **Closed-form conserved quantities that we wouldn’t know about** Although the most impactful conservation laws are likely to have already been discovered, Science is vast and always evolving. Moreover, symmetries and conservation laws form an integral part of Physics (for instance Nobel laureate Philip Anderson said “It is only slightly overstating the case to say that physics is the study of symmetry”). If recent results in various subfields are any indication, the discovery of new symmetries and conservation laws is still ongoing. For instance, it was recently discovered [1,2] that high concentrations of electron gas in nanodevices made of graphene behaved as a fluid due to having a conserved momentum as a subsystem in addition to the classical momentum conservation of the overall system. This allows electrons to go much faster than was expected. Other examples are many-body-localized systems with local conservation laws [3] and conformal symmetries in quantum field theory [4], which are less recent (1984) but very impactful.
>
> Finally, note that our framework allows for _approximately_ conserved but useful quantities. For instance, non-relativistic formulas of the energy are only approximately conserved, yet these would still likely help generalization for predicting systems of low velocities. Similar useful approximations to some more exact models could be valuable.
>
> **Visualizing Physics 101 data, predictions, and embeddings** We will add figures with examples of the Physics101 data and our model’s predictions to the appendix. For your reference, the images look like those in Figure 5 of [5]. Frames from one of our model’s generated sequences are included in Figure 1 of our main text; the bottom row is the initial sequence, and the top row is the final sequence generated after the inner optimization.
>
> We performed an analysis of the meta-learned embeddings as suggested by both you and reviewer A5RG; the following paragraph appears in our response to their review as well.
>
> We have generated visualizations of the “important” regions of several sequences in the test dataset according to the meta-learned embedding using Grad-CAM. This produces localization maps which highlight regions in each frame that most contribute to the magnitude of each of the eight dimensions of the embedding. In general, the behavior that we observe is structured and consistent across sequences with varying objects and ramp orientations. Notably, one dimension in particular consistently attends to the sliding object, following it down the ramp. Several of the other dimensions heavily attend to the edge of the ramp nearest the camera, while slightly attending to one of the two objects. We hypothesize that this behavior helps identify and conserve the orientation of the sequence, since half of the training and testing sequences are horizontally flipped. Two of the dimensions appear to consistently have near-uniform attention. In sequences where the human hand that releases the object is prominently visible, one particular dimension attends heavily to the hand, likely to enforce the presence and configuration of the hand in the generated frames.
>
> **Clarity** Thank you for your pointers on places where we should improve the clarity. Regarding tailoring in section 3.1, we will add a meta-tailoring diagram in the appendix and the test time pseudo-code in Algorithm 1 on the main text. Regarding the first paragraph of section 2, we will prune the unnecessary general notation as you suggest and replace it with intuitive descriptions. Yes, your understanding of the theory section is correct; we have a diagram showing how applying $\phi$ and then $\phi^{-1}$ lowers the dimensionality and visually shows how the space of possible outputs is reduced. We will add it to the paper.
>
> **Additional feedback**
> - In the theory, we could have used a fixed, non-linear feature $g(x)$ instead of $x$. However, we thought using $x$ matched physics better, where we have continuous trajectories, and was clearer for conveying our message. Its linear aspect is necessary to make the analysis. Note that this is a very common mechanism in ML theory to analyze a new behavior or mechanism [6,7,8,9].
> - In lines 65-66, $\phi$ corresponds to the embedding that must be conserved along a trajectory. $G$ is the generalization gap, i.e. the difference between expected test loss and empirical training loss (see the formula just above line 66). In general, we want the expected test loss to be small, which usually implies we want G to also be small. In the theory section, we prove upper-bounds on G as a function of the dimensionality of the preimages of $\phi$.
> - The second diagram mentioned in the Clarity section of our response (above) will explain the difference between $d$ and $m$. We will also highlight the letters in equations (1) and (2) to make their differences more apparent.
> - Our code will be open-sourced.
>
> [1] Hydrodynamics of electrons in graphene; Lucas & Fong ‘17
>
> [2] Higher-Than-Ballistic Conduction of Viscous Electron Flows; Guo et al. ‘17
>
> [3] Local Conservation Laws and the Structure of the Many-Body Localized States; Serbyn ‘13
>
> [4] Infinite conformal symmetry in two-dimensional quantum field theory; Belavin ‘84
>
> [5] http://phys101.csail.mit.edu/papers/phys101_bmvc.pdf
>
> [6] Adversarially Robust Generalization Requires More Data; Schmidt et al. ‘18
>
> [7] Unlabeled data improves adversarial robustness; Carmon et al. ‘19
>
> [8] Sharp Statistical Guarantees for Adversarially Robust Gaussian Classification; Dan et al. ‘20
>
> [9] Improving Adversarial Robustness via Unlabeled Out-of-Domain Data; Deng et al. ‘21

---

> > ### Comment · Reviewer_HJHv · 2021-08-18
> > **Reply to rebuttal**
> >
> > I thank the reviewers for their detailed comments.
> > In particular for the references regarding 'recent' discoveries of conservation laws. The simplification of the algorithm in the symbolic approach is also a good improvement.
> > However I am still struggling to be strongly convinced by the empirical results for reasons that also other reviewers highlighted. I still believe that the lack of clarity at some places in the paper is quite detrimental for the overall quality of the paper. I hope that the authors will remedy this issue.
> > For these reasons I stand by my original rating and encourage the authors to improve their submission following the reviewers recommendations.

---

### Official Review · Reviewer_3378 · 2021-07-16

**Rating:** 7
**Confidence:** 4

**Summary:**

**Summary**:
This paper is concerned with the problem of automatic symmetry discovery, by meta-learning conserved quantities that are useful for predictions. They propose Noether Networks, where the learned symmetries take the form of a meta-learned conservative loss optimised inside the prediction function during test time. They demonstrate experimentally that the proposed approach can improve prediction quality and discover useful inductive biases in sequential problems.

**Contributions**:
- This paper proposes a new algorithm that automatically learns its inductive biases in the form of a meta-learned conservative loss optimised inside the prediction function.
- (In a simplified setting) They theoretically show when and why enforcing conservative laws brings benefits.
- They demonstrate through experiments that the proposed algorithm can recover known physical laws, and improve prediction performance for video data.


**Ethical Concerns:**

I have no ethical concerns regarding this work.

**Limitations And Societal Impact:**

**Limitations**:
I think the authors already address some of their limitations. One suggestion would be: A theoretical or empirical analysis of why the learned conservation loss must correspond to symmetries rather than a generic form of regularisation.

**Societal Impact**:
I think the authors already have a good discussion of potential societal impact.


**Main Review:**

**Originality**:
Firstly, meta-learning symmetries is a relatively new and under-explored problem, and this paper offers a simple yet effective framework. Secondly, the idea of meta-learning conservation loss optimised inside the prediction function during test time has not been explored before, and the authors show empirically that this is actually a viable direction. Thirdly, this paper demonstrates the success of meta-learning continuous symmetries and learning symmetries that are useful even in noisy or partial observation settings, which I think is a big obstacle for wider applications of symmetries.

**Quality**:
Although under a simplified setting, the theoretical analysis in section 2 still offers some insight on when and why enforcing conservation laws is beneficial. The conclusion is sensible and expected, but I did not check the mathematical statements rigorously.

The methodology is simple and clean. I’d be curious to hear the following because I think it is less clear in the manuscript:
- When a conservation loss is learned, does it always correspond to certain continuous symmetries? (the opposite of Noether’s theorem) How does it compare to learning a regulariser in general?
- To enforce the conservation laws, why did the authors choose tailoring and optimise conversion loss during test time. What is the motivation behind this choice?

The experiments are all sensible and well-conducted. But I would be more convinced if the authors can show on video data what would happen when the unsupervised loss does not take the form of conservation loss. Is it obviously inferior compared to a conservation loss?

**Clarity**:
The paper is well written and it is enjoyable to read. The methodology and theoretical analysis are clean and well-paced.

**Significance**:
This paper presents a simple yet effective framework to discover continuous symmetries in the form of a meta-learned conservation loss. It has been demonstrated to be effective for discovering continuous symmetries, and can be applied to challenging settings where the conservation law is not exact due to noise, or only partial information is available. I believe it is an interesting direction for future exploration.


**Time Spent Reviewing:**

5

---

> ### Author Response · Authors · 2021-08-11
> **Response to Reviewer 3378**
>
> Thank you for your detailed review and positive comments. We answer your questions below:
>
> **Does a conservation law always correspond to a continuous symmetry?** The conditions for the inverse of Noether’s theorem are somewhat stricter; i.e. conservation laws are slightly more general. However, under the assumptions of Noether’s theorem conservation laws often have a corresponding symmetry; you can see [1] for a discussion. Note, however, that Noether’s theorem assumes the system of interest can be modeled only with a Lagrangian, which is not true when it is dissipative or partially observed.
>
> **How does our procedure compare to learning a regulariser in general?** Many regularisers (like L1,L2 losses) act on the weights. However, learning a regulariser there requires a network of size proportional to the network we want to regularize, which defeats the purpose of regularization. We could also learn a generic loss function that took in the entire sequence; however, the space of such functions is much larger and notably varies with the number of time-steps T, making both learning and implementation harder. We propose to include the meta-inductive bias of conservation, which applies to many physical settings. We will add this discussion to the main text. Finally, many regularisers take the form of auxiliary losses, rather than tailoring losses, which brings us to your next question.
>
> **Why use tailoring to enforce the conservations?** Tailoring is a recent idea to more properly impose known inductive biases by optimizing them within the prediction function. The classic alternative would have been to add our meta-learned conservation loss as an auxiliary loss (i.e. optimize $L_{task}+L_{cons}$ without any inner loop). This would have had two issues:
> - Auxiliary losses would optimise the conservation loss for the training points, but not at test time. This would result in the meta-learned embedding not being conserved at test time, resulting in a generalization gap on this loss. Intuitively, in Machine Learning we usually want training to mimic testing; meta-tailoring does this by always optimizing the loss within the prediction function, auxiliary losses don’t do this.
> - Meta-learning an auxiliary loss would have been much harder. This is because in auxiliary losses we pool the losses for all examples and all epochs and we only know their effect at validation/test time. Some works [2,3] have meta-learned data augmentation strategies or hyper-parameters in image classification using implicit gradients to find their effect on the final weights at the end of training. However, this is much less direct than with tailoring losses, where we directly use the update on the same sample just after using the loss. It would be interesting to see whether some of the cited previous works could be improved by prediction-time enforcing, instead of training behavior.
>
> [1] https://physics.stackexchange.com/a/28140/171648
>
> [2] Optimizing Millions of Hyperparameters by Implicit Differentiation; Lorraine et al. ‘19
>
> [3] Teaching with Commentaries; Raghu et al. ‘20

---

> > ### Comment · Reviewer_3378 · 2021-08-23
> > **I stand by my original rating**
> >
> > Thanks for providing the explanation and extended discussion. Based on these answers, I think their choices are sensible and justified. I will therefore stand by my original rating.

---

### Official Review · Reviewer_A5RG · 2021-07-16

**Rating:** 7
**Confidence:** 3

**Summary:**

The authors present a meta-learning approach for identifying conserved quantities in time and leveraging them in time series predictions. They demonstrate the utility of their approach on a series of physical systems and real-world settings.

**Limitations And Societal Impact:**

Potential limitations and societal impact have been sufficiently addressed.

**Main Review:**

The idea of formulating a loss based on time conservation is very interesting in the context of time series, as it allows to account for additional symmetries in a data driven fashion. Identifying those symmetries directly can be a significantly harder task using the original data.

The experiments conducted show that their approach can successfully identify existing conserved quantities and improve predictive performance in physical setups and video prediction tasks. Although the improvements in the latter tasks are minor, they still serve as a demonstration of the algorithms utility, since in these cases the functional form of the conservation law was learned in a completely data driven fashion. As such, an analysis of the learned function (e.g. the filters of the CNN) would be of high interest to the reader, which is currently missing from the text. Another point which would warrant a more detailed discussion is how learned symmetries can be applied to the initial model. Currently this is done via a single gradient update of the weights, however this might not be the best strategy in all scenarios (e.g. due to the strong non-linear dependence of a function on the weights). Evidence for this can be seen in the pendulum control experiments, where it proved to be more beneficial to adjust the activations.

The algorithm, experiments and results are described in a clear fashion and theoretical background is provided on when it can be beneficial to account for conservation laws.

On the whole, the approach presented in the paper can be considered an interesting step towards learning conserved quantities and has the potential to serve as a conceptual basis for future developments, although a more in-depth analysis would be desirable.

**Time Spent Reviewing:**

3

---

> ### Author Response · Authors · 2021-08-11
> **Response to Reviewer A5RG**
>
> Thank you for your encouraging comments regarding the interest of the proposed idea. We respond to your two main comments:
>
> **Encoding biases on the conservation loss and analyzing them** Understanding how to parameterize the encoding of the neural conservation losses and what they learn is of great interest. The class of networks that check for conditions on the final output could look very different from those that directly generate such an output. This would open a new avenue for architecture discovery. With respect to interpreting their behavior, we analyzed the PCA of the embeddings and found the biggest component depended on whether the ramp decline faced right or left, which affects a large number of pixels as well as the velocity of the object. We will add these visualizations to the appendix.
>
> **Embedding CNN filter analysis** We performed an analysis of the meta-learned embeddings as suggested by both you and reviewer HJHv; the following paragraph appears in our response to their review as well.
>
> We have generated visualizations of the “important” regions of several sequences in the test dataset according to the meta-learned embedding using Grad-CAM. This produces localization maps which highlight regions in each frame that most contribute to the magnitude of each of the eight dimensions of the embedding. In general, the behavior that we observe is structured and consistent across sequences with varying objects and ramp orientations. Notably, one dimension in particular consistently attends to the sliding object, following it down the ramp. Several of the other dimensions heavily attend to the edge of the ramp nearest the camera, while slightly attending to one of the two objects. We hypothesize that this behavior helps identify and conserve the orientation of the sequence, since half of the training and testing sequences are horizontally flipped. Two of the dimensions appear to consistently have near-uniform attention. In sequences where the human hand that releases the object is prominently visible, one particular dimension attends heavily to the hand, likely to enforce the presence and configuration of the hand in the generated frames.
>
> **Updating different parts of a network** As you mention, we found that directly updating the weights may not always be the best implementation in practice. As you also mention this is mostly about the non-linear relation between weights and final outputs, which get compounded due to the depth of the computation graph (both because of the number of time-steps and because of the number of inner optimization steps). We found that directly optimizing the final outputs of the LSTM leads to a more stable optimization. These activations still have a non-linear relationship with the output, due to the non-linear decoder, but are more stable because they do not go through multiple time-steps within the LSTM. We will explain this in greater detail in the final version.

---

> > ### Comment · Reviewer_A5RG · 2021-08-17
> > **Updated rating**
> >
> > I thank the authors for their clarifying comments. I believe that the overall quality of the paper will benefit from the additional analysis and the textual changes suggested in the responses to the other reviewers. I have changed my rating accordingly.

---

### Official Review · Reviewer_eyYP · 2021-07-17

**Rating:** 7
**Confidence:** 3

**Summary:**


**Summary and Contributions**
The paper proposes to approximately conserve desired quantities in sequential problems, such as when predicting the physical trajectory of an object. They argue intuitively that learning the right conserved quantities would be a useful inductive bias because symmetries (which are typical inductive biases) are often related to a conserved quantity through Noether Theorem. The proposed method meta-learns the quantity to be conserved from the data.

On the theoretical side, they study a specific setting where they prove an upper bound on the generalization gap which can be decreased when conservating some quantity. This suggests that learning models that conserve the right quantities could improve generalization.

They show in simple evaluation settings that the proposed method approximately learns the correct quantity to conserve and that conserving this quantity can help generalization.

**Limitations And Societal Impact:**

Limitations and societal impact are adequately discussed.

**Main Review:**

**Strengths**
- **Nice and interesting idea that will be of interest to the community**.The idea and argument for learning the quantities to conserve instead of the symmetries is very compelling and will be of interest to the symmetry/physics/ML subcommunity. I find the explanation provided on lines 36-38 particularly compelling: “whereas symmetries [… ] perturbating the actual data, conserved quantities […] only the true data”.
- **Impressive results: recovering the hamiltonian** I’m impressed by the fact that you essentially recover the mathematical formulation of the correct hamiltonian even in noisy environments. The proposed method that achieves this is interesting and a valuable contribution.
- **Novelty** to my knowledge using conservation of a quantity as an inductive bias is surprisingly novel
- all theoretical results seem correct to me.

**Weaknesses**
- **Somewhat misleading characterization of your theory in abstract and introduction** The theory sounds much stronger than it actually is. Line 10-11 (abstract): “We show theoretically […] that Noether Networks improve prediction quality”, and lines 55-56 (intro) “We theoretically characterize […] lowering the generalization gap”.  Both are a mischaracterization of the theoretical results which you provide for 2 reasons. First, the theory only holds in a constrained case where the preimages of the conserved law form parallel affine subspaces and the model is constrained to translating the input. Second, even under the aforementioned constrained, your results do not prove anything about the generalization gap, only upper bounds which could be very loose. If you wanted to use bounds to prove that models that conserve quantities have smaller generalization gaps than those which don’t, you would need to provide a LOWER bound for theorem 1 and show that this lower bound is larger than the upper bound from Theorem 2. To be clear, I do not think that section 2 is problematic, but the language in the abstract and intro should not suggest that you prove any strong theoretical results (use words such as “suggest”, “could”, “restricted setting”).
- **Could be more clear** I find that many parts of the paper were not very clear (at least not until reading all the paper). E.g,
    - Figure 1 is probably your most important figure in that most people will try to understand your method from this. I did not understand it at all before reading the paper, and even now I find it hard to understand. I would especially suggest (1) making it clear what is the difference between the top and the bottom sequences; and (2) using the caption to explain in more detail what is seen in the figure.
    - The first paragraph on page 3 is very hard to understand due to having to parse all that notation at once. I would suggest trying to explain with English words what is happening. For example, the fact that you are assuming that input space can be partitioned in subspaces in which the desired quantity is conserved (i.e., the preimages) and that these partitions are parallel affine subspaces of dimension m. Diluting the notation with some intuition / English explanations would really help the reader.
    - I find the explanation of (meta)-tailoring in the text hard to understand. Maybe having a figure would help. Your algorithm does somewhat help but it doesn't show what happens at test time. Part of the issue is that you use the language of meta-learning but say it's different because you use it in a single task setting. Maybe you could stick with standard meta-learning and say that you treat every sequence as a single "subdataset". This is for example what people do when talking about neural processes.
    - when reading the intro and abstract I originally thought that you would use the conserved quantities to learn the symmetries. But the entire paper has nothing to do with symmetries: you simply hypothesize (using the intuition behind Noether theorem) that conserving quantities is a useful inductive bias because that's related to symmetries which have been shown to be a useful inductive bias. I think it would be useful to make it explicit right after your statement of Noether's theorem that you do not do anything with symmetries.
- **Experimental results are hard to judge** besides for the first experiment, the experimental results are hard to judge as (1) the only baseline doesn't even try to add some inductive bias; (2) there are no standard benchmarks for these types of results; (3) on the provided line plots the results seem quite similar and the lack of standard errors make it hard to judge if these results are significant.

**Questions**
- I do not understand how you avoid learning a trivial/constant value to conserve (i.e., problem 2 line 121). In particular, as you are performing training and meta-training on the training set, I do not understand what is the advantage of learning a useful conservation law (I agree that it would be the case if you used a validation set). Can you please clarify?
- To obtain symbolic formulas of the Hamiltonian, I don’t really understand what you do once you have 20 candidates. Do you train all of the 20, and then select the best one on the training set?

**Suggestions / specific issues**
- Please remove the main text from the supplementary material.
- In the future, please do not remove numbered lines for reviewing (the first paragraph of page 3 has no numbers)
- For theorem 1 and theorem 2, the loss will often be the cross-entropy and so $C= \infty$, right? If so, it should at least be pointed out that the bounds are vacuous for log loss.
- Please be consistent with your notation. For example, $\phi$ in section 2 is a function, but a parameter for the (same?) function $g_\phi$ in section 3. As another example you are using $f_{theta}(x)$ in section 2, $f_{theta}(x,t)$ in section 3.3 and then back to $f_{theta}(x)$ in Algorithm 1.
- In section 4.2. please mention the incurred computational overhead. In particular, it seems to be very large for finding symbolic formulas, right?
- Please add in the main text that you had to remove some datasets from Greydanus because the DSL couldn’t find the form of the symbolic formula.
- Lines 515-517 of the appendices show an important advantage of your method, which I would briefly mention in the main text.
- In the proof of theorem 2, I would suggest stopping after line 501 and simply saying “the rest of the proof is the same as for theorem 1 but uses $\mathcal{}(r,\mathcal{Z}) \leq (2R \sqrt{m}/r)^m$". The redundancy of the current proof forces the reader to reread that part of the proof to understand how the result is different.

**Minor Suggestions**
- Line 30 / 33 / 114: i.e -> i.e.,
- Line 59: improve -> improving
- Top Paragraph of page 3: In the first occurrence of $i$ and $n$ please explain that it refers to the dataset/samples.
- Line 100: ensure satisfies -> ensure that it satisfies
- Line 100: bias -> unsupervised loss
- Line 136: described -> describe
- Line 242: cos q -> cos(q)
- Appcs line 457: remove “at”
- Appcs line 460: t he -> the
- Appcs equation 19: this is exactly the same as equation 15, I’m not sure what you wanted to add.

**Time Spent Reviewing:**

8 hours

---

> ### Author Response · Authors · 2021-08-11
> **Response to Reviewer eyYP**
>
> Thank you for your very detailed review! Your comments helped us strengthen and clarify the paper both in the main text and the appendix.
>
> **Properly characterizing the theory** We will modify the description of our theoretical results in the abstract and introduction to better reflect the strength of the results because, as you say, fully proving an improvement would require finding a complementary lower bound. Although you make it clear in your review that you are not critiquing the strength of the theory section, we would like to note that the use of linear or translation models to understand a phenomenon is a common technique in ML theory [1,2,3,4].
>
> **Clarity of Figure 1** We have reworked Figure 1 according to your suggestions, adding an “Initial Prediction” label to the bottom generated sequence and a “Final Prediction” label to the top sequence. We have also modified the bussing in and out of the Noether module to make it more clear that there are $T+1$ inputs (the embeddings) and a single output ($f_{\theta_{x_0}}$). We have also color-coded the two sequential prediction models, $f_{\theta}$ and $f_{\theta_{x_0}}$. Additionally, we have modified the caption to include a more detailed description of the flow of information in the diagram.
>
> **Improving the clarity of theory section** Multiple reviewers critiqued the clarity of the first paragraph of page 3; diluting the notation with plain English words is a great suggestion, which we will apply. We have created a diagram that intuitively shows how applying $\phi$ and then $\phi^{-1}$ lowers the dimensionality and reduces the space of possible outputs. We will add it to the paper.
>
> **Improving the description of meta-tailoring** By creating a different description of meta-tailoring vs. meta-learning we wanted to avoid the confusion of needing different tasks or distributions, since we are still in the standard ML setting. However, as you suggest, we can better align our notation and language with that of meta-learning to improve clarity, as well as add a ‘test-time’ algorithm to Algorithm 1. Thanks for the suggestions!
>
> **Why trivial quantities are avoided.** Both the prediction network and the meta-learned conservation embedding are trained end-to-end to minimize the task loss after the conservation update. However, for the meta-learned embedding this is equivalent to encouraging a big decrease in supervised loss after the update than before the update. This can be seen by looking at the gradient of the difference between the task loss before and after the update:
>
> $$\mathcal{L} (f_{\theta (\phi, x_0)} (x_0), x_{1:T}) - \mathcal{L} (f_{\theta} (x_0), x_{1:T}).$$
>
> Here, $\theta(\phi, x_0)$ denotes the model parameters after the inner loop update which conserves the embedding $g_\phi$ for input $x_0$. Notice that the second loss does not depend on the embedding (since $\theta$ has not been tailored using the embedding  $g_\phi$), thus its gradient is the same as that of the first term $\mathcal{L} (f_{\theta(\phi, x_0)} (x_0), x_{1:T})$ which is the one we optimize. A trivially conserved value (say $g_\phi=0$ independent of $\theta$ or $x$) will not produce any improvement on the supervised loss after its conservation is encouraged. Therefore, the embedding will seek to be more useful. We will add this discussion to the main text, since, in hindsight, it is relevant.
>
> **Final step in the Hamiltonian experiments** Yes, exactly; we train the final 20 candidates on the training set, using each loss in the inner loop, for next-step predictions (here $T=1$ to match the I/O spec of the baselines from the HNN paper). We then pick these trained weights and use the Runge-Kutta integrator (also used by the other baseline) to predict the trajectories of the appropriate length and pick the Noether loss that results in these trajectories having lower task loss. Notice that these trajectories still belong to the training set, as using the test set would not be appropriate. For the real pendulum, with only partial conservation, the number of steps and learning rate are also important as we only want to partially conserve the loss. These parameters are searched in the exact same way as the meta-learned losses, and also on the training data. We will make it clearer how this search is performed and that all computations use only the training data.
>
> **Shorter comments**
> - We will add a comment on the introduction clarifying the role of symmetries.
> - We will add error bars on the video experiments to help with judging their significance.
> - You’re correct that for the cross-entropy loss $C=\infty$. However, note that we’re in the regression setting where $x,y\in\mathbb{R}^d$, whereas cross-entropy is mostly used in classification where $y$ is a discrete variable. We will add a note saying that for those losses with $C=\infty$, the bound is vacuous.
> Good point about the notation change on $\phi$, we’ll fix this!
> - The computational overhead is indeed large for the formulas, where the search is discrete. We will make this clearer. Note that the discrete search only plays a part at training time; at test time we will have a slight increase because we are optimizing an unsupervised loss within the prediction function, but not nearly as big.
> - For conciseness, we note we will apply all your other suggestions not noted here. Thank you for looking at the main text and appendix in such great detail!
>
> [1] Adversarially Robust Generalization Requires More Data; Schmidt et al. ‘18
>
> [2] Unlabeled data improves adversarial robustness; Carmon et al. ‘19
>
> [3] Sharp Statistical Guarantees for Adversarially Robust Gaussian Classification; Dan et al. ‘20
>
> [4] Improving Adversarial Robustness via Unlabeled Out-of-Domain Data; Deng et al. ‘21

---

> > ### Comment · Reviewer_eyYP · 2021-08-18
> > **Thank you, I stand by my original review.**
> >
> > I've read in detail all the reviews and your answers. Thank you for all your answers.
> >
> > I stand by my original review: this is a novel and interesting idea that will be of interest to the community. As pointed out by Reviewer HJHv
> >  and me, the original manuscript nevertheless suffers from:
> > (1) lack of clarity at some places in the paper;
> > (2) experimental results are hard to judge.
> >
> > It seems that (1) will be improved. Error bars will marginally help (2), but the experimental results will still be a little hard to judge. I hope that this community quickly comes up with standard benchmarks and baselines for comparing methods.

---

### Decision · Program_Chairs · 2021-09-27

**Decision:**

Accept (Poster)

**Comment:**

All reviewers were happy in the end to accept this paper: I vote to accept. The biggest changes the reviewers wanted to see in the camera ready include: (i) improving the clarity, (ii) additional visualizations/experiments. Making these changes as well as others suggested by the reviewers will make a good paper a great one and a welcome addition to the conference.